# High Performance Differentially Private Fine-Tuning using Dataset Distillation

## Abstract

Differentially Private Stochastic Gradient Descent (DP-SGD), which iteratively perturbs clipped per-sample gradients and tracks the cumulative privacy risk using composition accounting, has become a cornerstone in private deep learning. Despite its versatility, DP-SGD in practice faces several limitations. It is constrained by the number of gradient iterations permissible under a limited privacy budget, and is restricted by incompatibilities with common deep learning techniques like ensembling and BatchNorm, and typically produces only a single trained model. In this work, we propose an algorithm for generating a differentially-private (DP) synthetic version of a sensitive dataset. This allows the synthetic dataset to be distributed and postprocessed freely without additional privacy loss, giving more flexibility than DP-SGD. Building on dataset distillation—by producing compact synthetic datasets that preserve downstream performance— we introduce SPS (Summarize–Privatize–Synthesize) and its enhanced variant SPS+. In contrast to prior works, SPS is, to our knowledge, the first alternative to DP-SGD that attains higher accuracy on image-classification tasks. Concretely, on CIFAR 10 / CIFAR 100 with privacy budget $\epsilon = 1$, SPS+ achieves **96.2**/**76.6**% top-1 accuracy, outperforming state-of-the-art (SOTA) DP-SGD results (94.8/70.3%).

## 1 Introduction

Deep learning is effective in predictive performance and efficient in implementation when trained on non-sensitive data, but it falls short when datasets contain private information. Numerous practical attacks can efficiently reconstruct training data from trained models in the absence of proper protections Zhu et al. (2019a); Buzaglo et al. (2023). To provably mitigate adversarial inference, a widely adopted framework is *Differential Privacy* (DP) Dwork (2006), which quantifies and allows one to control the advantage of any adversary in determining the membership of an individual data record. Deep Learning with DP constraints has long proved challenging: not only do practitioners pay a price in terms of accuracy, the most common and performant method for differentially private learning: Differentially-Private SGD (DP-SGD) Abadi et al. (2016), comes at a computational and practical cost. DP-SGD faces practical limitations: restricted iterations due to composition budgets, incompatibility with BatchNorm (Ioffe, 2015) and ensembling Ganaie et al. (2022), and expensive per-sample gradient computation. Furthermore, if the data is to be used again in the future, one must retrain old models so account for the additional privacy cost of training the new ones.

Alternatively, one can *privatize* a dataset by generating a synthetic version that satisfies DP guarantees. This approach offers far greater flexibility than DP-SGD: the synthetic corpus can be publicly released, reused to train multiple models, and probed with data-attribution–based explainability methods—capabilities that are typically infeasible under standard DP-SGD. An especially compelling use case arises when sensitive data are siloed across multiple databases, each with its own privacy restrictions. If each curator independently releases a DP-synthetic dataset, these can be aggregated into a global public corpus, allowing mutual benefit without leaking private information. Despite this promise, generation-based approaches have historically lagged gradient-based training in accuracy, limiting their practicality.

Meanwhile, recent progress in dataset distillation—the study of crafting small synthetic datasets that train to high accuracy—offers guidance for building high-performance synthetic data. For example, the D3S algorithm (Loo et al., 2024), which matches intermediate-layer feature statistics, is among

the first methods to scale to larger models. Although recent work has combined dataset-distillation techniques with DP (Vinaroz & Park, 2024), its performance remains below that of DP-SGD. In this work, we show for the first time that distillation-based approaches to privacy can match or exceed the accuracy of DP-SGD on private image classification while providing greater flexibility. In particular, our contribution can be summarized as follows:

1. **Present SPS**, a differentially private dataset distillation algorithm that adapts D3S to work with public pre-trained models while privatizing intermediate activation statistics;

2. **Develop multistage clipping and grouped pseudo-classes** techniques that significantly improve performance in high-privacy regimes, yielding the enhanced SPS+ algorithm;

3. **Demonstrate competitive performance** with DP-SGD on CIFAR-10/100 classification, becoming the **first** generation-based method to match gradient-based approaches;

4. **Show practical advantages** of data-based privacy including support for model ensembling, federated learning, and continual learning without additional privacy cost.

## 2 BACKGROUND

### 2.1 DIFFERENTIAL PRIVACY

DP defines the privacy risk by measuring the worst-case (maximal) divergence between the output distributions of some mechanism $\mathcal{M}$ on two *adjacent* datasets differing in a single data point. In the following, we formally introduce a well-known variant of DP, Rényi Differential Privacy (RDP).

**Definition 2.1** ( $(\alpha, \epsilon(\alpha))-$Rényi Differential Privacy Mironov (2017b))**.** *Given a universe $\mathcal{X}$, we say that two datasets $X, X' \subseteq \mathcal{X}^*$ are adjacent, denoted as $X \sim X'$, if $X = X' \cup \{x\}$ or $X' = X \cup \{x\}$ for some additional datapoint $x \in \mathcal{X}$. A randomized algorithm $\mathcal{M}$ satisfies $(\alpha, \epsilon(\alpha))$-Rényi Differential Privacy (RDP), $\alpha > 1$, if for any pair of adjacent datasets $X \sim X'$, $D_\alpha(\mathbb{P}_{\mathcal{M}(X)} \| \mathbb{P}_{\mathcal{M}(X')}) \leq \epsilon(\alpha)$. Here, $\mathbb{P}_{\mathcal{M}(X)}$ and $\mathbb{P}_{\mathcal{M}(X')}$ represent the distributions of $\mathcal{M}(X)$ and $\mathcal{M}(X')$, respectively, and*

$$D_\alpha(\mathsf{P} \| \mathsf{Q}) = \frac{1}{\alpha - 1} \log \int \mathsf{q}(o) (\frac{\mathsf{p}(o)}{\mathsf{q}(o)})^\alpha \, do, \tag{1}$$

*represents $\alpha$-Rényi Divergence between two distributions $\mathsf{P}$ and $\mathsf{Q}$ whose density functions are $\mathsf{p}$ and $\mathsf{q}$, respectively.*

For a given $\alpha$, a smaller $\epsilon(\alpha)$ implies a more significant challenge for an adversary to distinguish the participation of an arbitrarily-selected data point, and the mechanism $\mathcal{M}$ preserves privacy better. RDP can also be converted back into the classic approximate $(\epsilon, \delta)$ DP (Mironov, 2017a; Canonne et al., 2020). We provide a full description in section C.

In many applications, multiple accesses to a sensitive dataset are required, and each additional release increases the potential privacy leakage. RDP can be used to elegantly handle the composition of privacy leakage, as given by Lemma 2.2

**Lemma 2.2** (Composition of RDP (Proposition 1 of Mironov (2017b)))**.** *Let $f : \mathcal{D} \mapsto \mathcal{R}_1$ be $(\alpha, \epsilon_1)$-RDP and $g : \mathcal{R}_1 \times \mathcal{D} \mapsto \mathcal{R}_2$ be $(\alpha, \epsilon_2)$-RDP, then the mechanism defined as $(X, Y)$, where $X \sim f(D)$ and $Y \sim g(X, D)$, satisfies $(\alpha, \epsilon_1 + \epsilon_2)$-RDP.*

Lemma 2.2 suggests that the security parameter $\epsilon(\alpha)$ in RDP increases *linearly* under composition. Most existing DP composition methods, including RDP (Lemma 2.2) and more refined techniques such as $f$-DP Dong et al. (2022) and characteristic function-based approaches Zhu et al. (2022), need to consider the *worst-case* scenario for each composed mechanism. Consequently, they are unable to handle *unbounded* composition. In the context of machine learning, this limitation implies that a fixed privacy budget *cannot* support training an unlimited number of privatized model using conventional composition techniques. This motivates the use of *private synthetic data generation* as a means to enable unrestricted private data utilization.

Another concrete application of composition is DP-SGD Abadi et al. (2016). It interprets a standard $T$-iteration SGD process as a sequence of $T$ adaptively composed single-iteration gradient computations and updates, and clips and adds noise to gradients to ensure DP-guarantees under $T$ compositions. Similarly, the iteration number $T$ must be bounded.

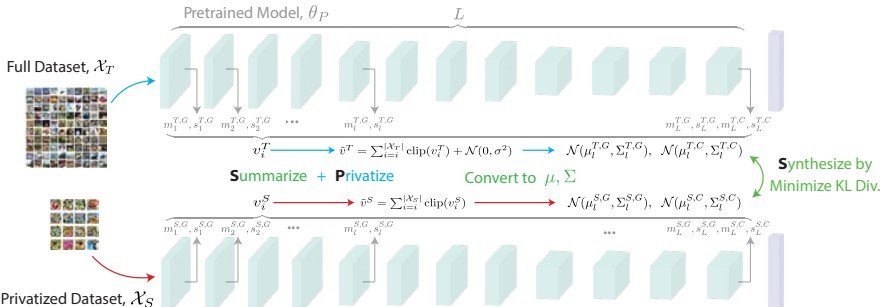

Figure 1: A schematic of the Summarize, Privatize, Synthesize (SPS) algorithm. SPS works by matching privatized summary statistics between the real dataset $\mathcal{X}_T$ and a synthetic one $\mathcal{X}_S$ using a pretrained model $\theta_P$.

## 2.2 DP Synthetic Data Generation

For private synthetic data generation, DP-SGD has been used to train generative models—including diffusion models (Ghalebikesabi et al., 2023; Dockhorn et al., 2022) and GANs (Xie et al., 2018)—but scaling these models remains challenging. Larger models typically require more iterations to converge, yet DP-SGD is limited by a finite composition budget; moreover, the injected noise grows with model dimensionality (Xiao et al., 2023), so bigger models endure heavier noise, further slowing convergence. Similar composition constraints also affect alternatives such as PATE (Private Aggregation of Teacher Ensembles) (Papernot et al., 2016), which requires access to additional unlabeled public data.

Other DP data-generation strategies exist: in images, one can leverage public diffusion models (Lin et al., 2024), or—closest to our approach—statistically match privatized mean embeddings (Harder et al., 2020). However, in the image domain, downstream classification performance still trails direct DP-SGD training: the best reported CIFAR-10 accuracy from synthetic data is $89.1\%$ (Lin et al., 2024), well below DP-SGD's $> 95\%$ (De et al., 2022). By contrast, in the text domain, fine-tuning on private synthetic data can outperform direct DP-SGD training (Xie et al., 2024; Amin et al., 2024). To our knowledge, our work is the *first* to achieve such parity in the image domain.

## 2.3 Dataset Distillation

Dataset distillation (DD) aims to produce compact synthetic datasets that preserve high downstream accuracy (Wang et al., 2018). Among major DD families—bilevel optimization (Wang et al., 2018; Loo et al., 2023), kernel-based methods (Nguyen et al., 2021a;b), trajectory matching (Cazenavette et al., 2022), and *activation–statistic matching* (e.g., D3S (Loo et al., 2024; Yin et al., 2023a))—the last is particularly well-suited to differential privacy. These methods align intermediate activation statistics (e.g., means and covariances at BatchNorm layers) between the full and distilled datasets.

Crucially, statistic-matching algorithms like D3S need to privatize only the *statistic-collection phase*, enabling DP via a single noise-addition step. By contrast, other DD approaches must privatize each optimization iteration, incurring costly iterative composition (Dwork, 2006). Prior private DD attempts, such as DP-KIP (Vinaroz & Park, 2024), attain only $58.7\%$ on CIFAR-10 at $(\epsilon, \delta) = (10, 10^{-5})$, far below DP-SGD's $> 93\%$ on the same benchmark (De et al., 2022).

## 3 Our Method

In this section, we present the **S**ummarize–**P**rivatize–**S**ynthesize (**SPS**) algorithm. Our method builds on the D3S dataset–distillation framework (Loo et al., 2024), but introduces substantial modifications to address the challenges of the differentially private (DP) setting. We first describe SPS in section 3, then introduce *multistage clipping* and *grouped pseudo-classes* in section 4, yielding the enhanced SPS+ algorithm, which performs significantly better on multiclass tasks.

## 3.1 Review: D3S Dataset Distillation

Dataset Distillation with Domain Shift (D3S) matches intermediate activation statistics between the full dataset $\mathcal{X}_T$ and the distilled set $\mathcal{X}_S$. Statistics are computed using a model $\theta_T$ trained on $\mathcal{X}_T$. Concretely, let the post–BatchNorm activation at layer $l$ for example $i$ be $z_{i,l} \in \mathbb{R}^{D_l \times H_l \times W_l}$. After averaging over spatial dimensions $(H_l, W_l)$, D3S forms per-layer means and covariances for the full and distilled sets,

$$\mu_l^T, \mu_l^S \in \mathbb{R}^{D_l}, \qquad \Sigma_l^T, \Sigma_l^S \in \mathbb{R}^{D_l \times D_l},$$

and seeks to match these statistics. Treating the distributions of $z_{i,l}$ as Gaussian, the objective minimizes the KL divergence between the corresponding normals together with a supervised loss on distilled labels $\mathcal{Y}_S$ (pre-assigned), evaluated by $\theta_T$:

$$\mathcal{L}_{\text{D3S}} = \sum_{l=1}^{L} D_{KL}\Big(\mathcal{N}(\mu_l^T, \Sigma_l^T) || \mathcal{N}(\mu_l^S, \Sigma_l^S)\Big) + \text{x-ent}\Big(f_{\theta_T}(\mathcal{X}_S), \mathcal{Y}_S\Big)$$

D3S further averages the loss across multiple trained models and employs an exponential–moving–average scheme to estimate $\mu_l^S$ and $\Sigma_l^S$ when the distilled batch size is large. Optimizing this loss yields synthetic images $\mathcal{X}_S$. To obtain soft targets, a knowledge–distillation–style procedure assigns soft labels to $\mathcal{X}_S$ using $\theta_T$.

However, the foregoing procedure is *not* private. First, the collection of $\mu_l^T$ and $\Sigma_l^T$ is non-private. More critically, the algorithm depends on a model $\theta_T$ trained on the full private dataset—both to guide image synthesis and to assign labels—so a private distillation method must eliminate (or carefully privatize) this reliance on $\theta_T$.

## 3.2 Adapting to Private Generation

Here we describe how we develop SPS, an algorithm similar to D3S which overcomes the aforementioned issues and works in the differentially private setting.

### 3.2.1 Removing the trained model

To address the issues above, we first remove reliance on the privately trained model $\theta_T$. The most straightforward remedy is to use a publicly pretrained model $\theta_P$ trained on a large non-sensitive corpus—a common practice in the DP literature (Mehta et al., 2022; De et al., 2022; Ganesh et al., 2023). However, substituting $\theta_T$ with $\theta_P$ introduces two challenges: **(i) missing class assignments** and **(ii) missing soft-label information**. In D3S, a cross-entropy term enforces class alignment for synthesized images, and soft targets are obtained via knowledge distillation from $\theta_T$—both unavailable without $\theta_T$.

To circumvent this, for a dataset with $C$ classes we collect $C$ sets of class-conditional statistics at a subset of layers $L_C \subseteq [L]$ (typically the last three). These statistics must be rich enough to capture distributional structure lost with hard labels, so we model *full* multivariate Gaussian summaries rather than matching means alone. During synthesis, images intended for class $c$ match the corresponding class-conditional statistics via a KL-divergence objective, while all synthetic data additionally match *global* (class-marginal) statistics averaged across classes. Let the global statistics be $(\mu_l^{T,G}, \Sigma_l^{T,G})$ and, for class $c$, the class-conditional statistics be $(\mu_l^{T,c}, \Sigma_l^{T,c})$; the resulting loss is:

$$\mathcal{L}_{\text{SPS}} = \sum_{l=1}^{L} D_{KL}\Big(\mathcal{N}(\mu_l^{T,G}, \Sigma_l^{T,G}) || \mathcal{N}(\mu_l^{S,G}, \Sigma_l^{S,G})\Big) + \lambda_C \sum_{c=1}^{C} \sum_{l \in L_C} D_{KL}\Big(\mathcal{N}(\mu_l^{T,c}, \Sigma_l^{T,c}) || \mathcal{N}(\mu_l^{S,c}, \Sigma_l^{S,c})\Big)$$

(2)

For some scalar hyperparameter $\lambda_C$, the assigned classes are then used for **hard labels** as opposed to soft ones. We show the importance of this change in section B.1. Additionally, we use use different dimensions $D_G$ and $D_C$ for global and class-specific statistics, typically choosing $D_G > D_C$. This is necessary because the class-specific statistics are subject to noise which is larger by a factor of $C$ when privatized, so we must keep their dimensionality smaller. To do this, for intermediate activation $z_{i,l} \in \mathbb{R}^{C_l \times H \times W}$ for data point $i$, we project it and apply a nonlinearity to get the embedding used for the global and class-specific embedding:

$$z_{i,l}^G = 2\sigma(M_l^G z_{i,l}) - 1 \in \mathbb{R}^{D_G \times H \times W}, \qquad z_{i,l}^C = 2\sigma(M_l^C z_{i,l}) - 1 \in \mathbb{R}^{D_C \times H \times W}$$

where $M_l^G$ and $M_l^C$ are random projection matrices of dimension $\mathbb{R}^{D_G \times D_l}$ and $\mathbb{R}^{D_C \times D_l}$, respectively, and $\sigma$ is the sigmoid function.

### 3.2.2 PRIVATIZING THE STATISTICS

The second change we need to make is to ensure collecting the global and class summarization are implemented in a privacy-preserving manner. For each datapoint $x_i$, we release the first and second moments given by:

$$m_{i,l}^G = \frac{1}{HW} \sum_{h=1}^{H} \sum_{w=1}^{W} z_{i,l}^G \in \mathbb{R}^{D_G}, \qquad s_{i,l}^G = \frac{1}{HW} \sum_{h=1}^{H} \sum_{w=1}^{W} z_{i,l}^G z_{i,l}^{\mathsf{T}G} \in \mathbb{R}^{D_G \times D_G}, \qquad 1 \le l \le L$$

$$m_{i,l}^c = \frac{\mathbb{1}_{(y_i=c)}}{HW} \sum_{h=1}^{H} \sum_{w=1}^{W} z_{i,l}^c \in \mathbb{R}^{D_C}, \qquad s_{i,l}^c = \frac{\mathbb{1}_{(y_i=c)}}{HW} \sum_{h=1}^{H} \sum_{w=1}^{W} z_{i,l}^c z_{i,l}^{\mathsf{T}c} \in \mathbb{R}^{D_C \times D_C}, \quad l \in L_C, 1 \le c \le C$$

$$(3)$$

It is noted that the second moment is a symmetric matrix and thus we only need to release the upper triangular part. Let $u = \mathrm{triu}(s)$ be the upper triangular part of $s$. Flattening and concatenating these into a single statistic vector $v_i$, we have, for each datapoint:

$$v_i = [\underbrace{m_{i,1}^G, u_{i,1}^G, \ldots m_{i,L}^G, u_{i,L}^G}_{v_i^G}, \underbrace{m_{i,1}^1, u_{i,1}^1, \ldots m_{i,L}^1, u_{i,L}^1}_{v_i^{c=1}} \ldots \underbrace{m_{i,1}^C, u_{i,1}^C, \ldots m_{i,L}^C, u_{i,L}^C}_{v_i^{c=C}}]$$

The total dimensionality of the released statistics $v_i$ is $d_{\mathrm{tot}} = L D_G^{\mathrm{layer}} + C |L_C| D_C^{\mathrm{layer}}$, where $D_G^{\mathrm{layer}} = D_G + \frac{D_G(D_G+1)}{2}$, corresponding to the dimensionality of a single layer's mean and covariance[1]. The same applies analogously for the per-class statistics captured by $D_C^{\mathrm{layer}}$. Thus, the overall dimensionality can be tuned by adjusting $D_G$, $D_C$, and $|L_C|$. Tuning the dimensionality of the privatized statistic is a key advantage of SPS of DP-SGD. Whereas DP-SGD is limited by the high dimensionality of gradients ($\sim 10^7$), by tuning $D_C$ and $D_G$ to be small, the dimensionality of SPS can be significantly smaller ($\sim 10^5$), thereby improving the SNR of the privatized statistic.

We are interested in releasing the aggregate sum of these statistic vectors: $\tilde{v} = \sum_{i=1}^{N} v_i$, computed over $N$ datapoints. This sum can be privatized using the Gaussian Mechanism (Dwork, 2006). Specifically, we clip each $v_i$ to satisfy $\|v_i\|_2 \le \|v\|_{\max} := K_{\mathrm{clip}} \sqrt{L D_G^{\mathrm{layer}} + |L_C| D_C^{\mathrm{layer}}}$, where $K_{\mathrm{clip}}$ is a positive constant (typically on the order of $10^{-1}$). We then add Gaussian noise with standard deviation $\sigma = b_0 \|v\|_{\max}$, where $b_0$ is chosen according to the privacy budget. The final released vector is

$$\tilde{v} = \sum_{i=1}^{N} \mathrm{clip}(v_i) + \mathcal{N}\left(0, b_0^2 \|v\|_{\max}^2 I\right), \qquad \mathrm{clip}(v_i) = v_i \cdot \min\left\{1, \frac{\|v\|_{\max}}{\|v_i\|_2}\right\}. \qquad (4)$$

Here, $\mathrm{clip}(\cdot)$ denotes the standard $\ell_2$ clipping function.

### 3.2.3 POST-PROCESSING THE STATISTIC VECTOR

After obtaining the full statistic vector $\tilde{v}$, we need to convert it back into means and covariances. Splitting $\tilde{v}$ back into summed first and second moments, we have $\tilde{m}_l^G, \tilde{u}_l^G$ for each layer and $\tilde{m}_l^c, \tilde{u}_l^c$ for each class and layer. Focusing on the global statistics, we process these into means and covariances as follows:

$$\mu_l^G = \frac{1}{N} \tilde{m}_l^G, \qquad \hat{\Sigma}_l^G = \frac{1}{N} \mathrm{triu}^{-1}(\tilde{u}_l^G) - \mu_l^G \mu_l^{\mathsf{T}G}$$

Where $\mathrm{triu}^{-1}$ converts $\tilde{u}$ back into the symmetric matrix $\tilde{s}$. We additionally clip negative eigenvalues of $\hat{\Sigma}_l^G$ to produce $\Sigma_l^G$, which we use during optimization. Details are provided in section A.3

We apply a similar procedure for the class-specific statistics, instead normalizing by $\frac{N}{C}$. This yields our final set of statistics for matching. Once we have these statistics, we synthesize datapoints by iteratively optimizing eq. (2), initializing with random noise images in $\mathcal{X}_S$. We generate images until we have reached our desired dataset size $|\mathcal{X}_S|$. Full algorithm pseudocode is available in section A.1.

---

[1]The $\frac{D(D+1)}{2}$ term arises from the symmetry of second-moment matrices.

### 3.2.4 BETTER CLIPPING BY REDISTRIBUTING NOISE

The standard clip-and-add-noise procedure performs poorly because per-class statistics are normalized by $\frac{C}{N}$ while global statistics use $\frac{1}{N}$, making per-class statistics $C$ times more susceptible to noise. This, in turn, degrades class matching in synthesized images. We remedy this by upscaling the per-class statistics by $\sqrt{S}$, where $S = \frac{LD_G^{\text{layer}}}{|L_C|D_C^{\text{layer}}}$, and release $v_i = [v_i^G, \sqrt{S}v_i^1, \ldots, \sqrt{S}v_i^C]$. After adding noise, we divide by $\sqrt{S}$ when reconstructing the per-class statistics. This redistributes noise to impact the global parameters more, while keeping the same privacy cost $b_0$. Correspondingly, we clip according to $|v|_{\max} = K_{\text{clip}}\sqrt{LD_G^{\text{layer}} + S|L_C|D_G^{\text{layer}}} = K_{\text{clip}}\sqrt{2LD_G^{\text{layer}}}$.

### 3.2.5 BETTER OPTIMIZATION

Finally, we detail a few techniques used in optimization to improve performance.

**Smooth Activations.** We use SiLU activations instead of ReLU in pretrained models $\theta_P$ to facilitate optimization through the network. Smooth activations improve input optimization tasks like reconstruction attacks and adversarial robustness (Shahin Shamsabadi et al., 2023; Xie et al., 2020), making them well-suited for our image synthesis process.

**Sharpness Aware Minimization (SAM) at validation time.** We fine-tune on distilled datasets using the GSAM optimizer (Foret et al., 2021; Zhuang et al., 2022). This choice is motivated by the evidence that SAM-style methods improve generalization under label noise (Baek et al., 2024), which is pertinent here because privatization injects noise. Although SAM typically complicates DP training by requiring two gradient evaluations per step, our setting—training on privatized (DP) synthetic data—falls under the post-processing property (Dwork, 2006). Consequently, any downstream optimizer, including GSAM, can be used without incurring additional privacy cost.

## 4 SPS+: SPS WITH BETTER CLIPPING AND GROUPING

While SPS provides differential privacy, it performs poorly in the high-privacy regime. In particular, the per-class statistics $v_i^c$ accumulate noise at a rate of $O(C/N)$, which can become prohibitive in few-sample settings. To address this, we introduce two key enhancements: *multistage clipping* (MC) and *grouped pseudo-classes* (GPC), together yielding the improved SPS+ algorithm.

### 4.1 MULTISTAGE CLIPPING

In the basic SPS framework, a single measurement of $\tilde{v}$ must be privatized. Prior work on DP mean estimation (Bie et al., 2023) shows that multistage methods—which iteratively adjust the clipping radius and recenter around previous (biased) estimates—can empirically outperform single-shot estimation. Multistage clipping (MC) adapts this idea to SPS. Specifically, we begin with clipping center 0 and factor $K_{\text{clip}}^1$, producing an initial synthetic dataset $\mathcal{X}_S^1$. In the next stage, we recenter the clipping operation at the empirical means computed from $\mathcal{X}_S^1$, initialize the optimization with $\mathcal{X}_S^1$, adopt a new clipping factor $K_{\text{clip}}^2$, and jointly optimize $\mathcal{L}_{\text{SPS}}$ over both stages to obtain $\mathcal{X}_S^2$. This process is repeated for $M$ stages, each time re-centering, re-clipping, and re-initializing from the previous synthetic dataset. A full description of the modified measurement procedure is given in section A.6. The privacy guarantee follows directly from composition, resulting in $M$-fold DP.

### 4.2 GROUPED PSEUDOCLASSES

To address $O(C/N)$ noise rate, rather than directly matching $C$ estimates of $v^c$, each with noise rate $O(C/N)$, we proposed generating $P > C$ *pseudo-classes*, which are composed of random groups $N_{c/p} > 1$ real classes, and matching pseudoclasses statistics with each other. As a result, each class belongs to $\frac{PN_{c/p}}{C}$ pseudo-classes. Each pseudoclass' statistics estimate has a more favourable $O(\frac{C}{NN_{C/p}})$ noise rate, allowing better optimization of eq. (2). Importantly, this technique **only works due to dynamics of optimizing the loss function**, specifically the $\Sigma$ inversion in the KL-

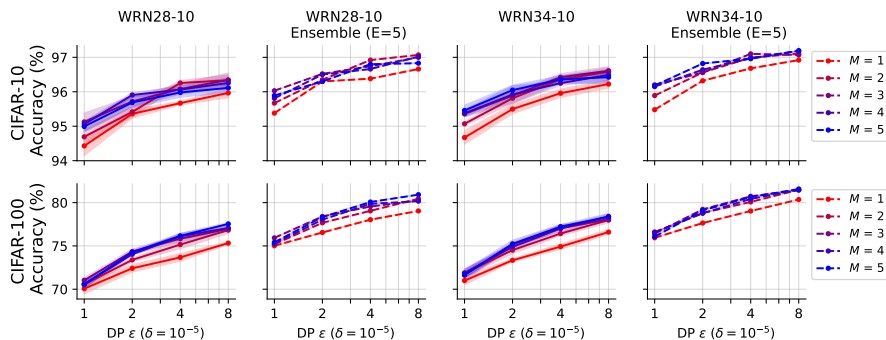

Figure 2: Accuracy of pretrained Wide ResNet models fine tuned on SPS+ synthesized images on CIFAR-10 and CIFAR-100. We evaluate $n = 5$ WRN28-10 and WRN34-10 models, and their respective ensembles, varying both privacy budget $\epsilon$, and the number of clipping stages $M$. Increasing $M$ can improve performance over $M = 1$.

Table 1: Accuracy of Differentially private fine-tuning of Wide ResNet Models on CIFAR-10 and CIFAR-100 at various privacy budgets $\epsilon$. Error bars are computed for $n = 5$ runs, ensembles use 5 models.

| Method | CIFAR-10 | | | | CIFAR-100 | | | |
|---|---|---|---|---|---|---|---|---|
| | $\epsilon = 1$ | $\epsilon = 2$ | $\epsilon = 4$ | $\epsilon = 8$ | $\epsilon = 1$ | $\epsilon = 2$ | $\epsilon = 4$ | $\epsilon = 8$ |
| DP-SGD (De et al., 2022) (WRN28-10) | $94.8 \pm 0.1$ | $95.4 \pm 0.2$ | $96.1 \pm 0.1$ | $96.6 \pm 0.1$ | $\underline{70.3 \pm 0.1}$ | $\underline{74.7 \pm 0.2}$ | $\underline{79.2 \pm 0.2}$ | $\mathbf{81.8 \pm 0.1}$ |
| SPS (WRN28-10) | $93.2 \pm 0.2$ | $94.6 \pm 0.2$ | $95.0 \pm 0.2$ | $95.9 \pm 0.1$ | $48.9 \pm 1.1$ | $54.0 \pm 1.0$ | $66.3 \pm 0.4$ | $70.7 \pm 0.4$ |
| SPS (WRN34-10) | $93.9 \pm 0.1$ | $94.9 \pm 0.2$ | $95.4 \pm 0.2$ | $96.1 \pm 0.1$ | $50.6 \pm 1.2$ | $53.7 \pm 0.9$ | $67.2 \pm 0.1$ | $72.2 \pm 0.3$ |
| SPS (WRN28-10 Ensemble) | $94.9$ | $95.8$ | $96.0$ | $96.5$ | $57.0$ | $59.7$ | $71.6$ | $74.9$ |
| SPS (WRN34-10 Ensemble) | $\underline{95.3}$ | $95.9$ | $\underline{96.3}$ | $\underline{96.8}$ | $59.2$ | $59.6$ | $71.8$ | $75.9$ |
| SPS+ (WRN28-10) | $95.1 \pm 0.3$ | $95.9 \pm 0.1$ | $96.3 \pm 0.1$ | $96.3 \pm 0.2$ | $71.0 \pm 0.3$ | $74.3 \pm 0.3$ | $76.2 \pm 0.3$ | $77.5 \pm 0.1$ |
| SPS+ (WRN34-10) | $95.5 \pm 0.1$ | $96.0 \pm 0.1$ | $96.4 \pm 0.1$ | $96.6 \pm 0.1$ | $71.9 \pm 0.5$ | $75.2 \pm 0.4$ | $77.2 \pm 0.2$ | $78.4 \pm 0.2$ |
| SPS+ (WRN28-10 Ensemble) | $96.0$ | $96.5$ | $96.9$ | $97.1$ | $75.9$ | $78.4$ | $80.1$ | $80.9$ |
| SPS+ (WRN34-10 Ensemble) | $\mathbf{96.2}$ | $\mathbf{96.8}$ | $\mathbf{97.1}$ | $\mathbf{97.2}$ | $\mathbf{76.6}$ | $\mathbf{79.2}$ | $\mathbf{80.7}$ | $\underline{81.6}$ |
| Private Evolution (Lin et al., 2024) | 89.13% ($\epsilon = 10$) | | | | - | | | |

divergence, and the eigenvalue clipping of $\Sigma$. This method **does not offer benefits for direct mean estimation**. More details are available in section A.5.

### 4.3 PRIVACY GUARANTEE FOR SPS

The privacy-sensitive step of SPS is the release of aggregation of individuals $v_i$s, which is given in eq. (4). This is directly private sum estimation using the Gaussian Mechanism. For SPS+, this is composed M times for each measurement step. We have the following privacy guarantee:

**Theorem 4.1** (Privacy of SPS). *The release of $\tilde{v}$ in eq. (4) for M models satisfies $(\alpha, \epsilon)$-RDP, where $\epsilon = \frac{M\alpha}{2b_0^2}$ for $\alpha > 1$. Proof. See section C.1* $\square$

This is a direct result of the M-fold composition of Gaussian Mechnisms under RDP. This can be converted to $(\epsilon, \delta)$-DP using Proposition 12 in Canonne et al. (2020), which is implemented in the RDP-account in Ahmed et al. (2025), which we use in the comparison with prior works.

## 5 RESULTS

### 5.1 FINE-TUNING WITH PUBLIC DATASETS

To validate our method, we first test it on fine-tuning publicly available pretrained datasets. Specifically, we take the task of generating synthetic private version of CIFAR-10 and CIFAR-100 (Krizhevsky, 2009) using a pretrained Wide ResNet 22-8 with SiLU activations trained on $32 \times 32$ resolution ImageNet (He et al., 2015; Zagoruyko & Komodakis, 2017; Deng et al., 2009), in line with prior work (De et al., 2022). Our privatized datasets are the same size as the original (50,000 images). During distillation, we vary the privacy budget $\epsilon \in \{1, 2, 4, 8\}$, with a fixed $\delta = 10^{-5}$. For SPS+, we keep $P = 20, 200$ pseudoclasses for CIFAR-10 and CIFAR-100, respectively, and vary $M$, the number of stages. Details on the choice of hyperparameters are given in section D.2.

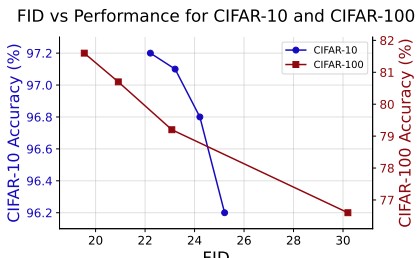

$\epsilon = 1$  FID = 42.00   $\epsilon = 2$  FID = 36.71   $\epsilon = 4$  FID = 32.13   $\epsilon = 8$  FID = 24.36

Figure 4: Visualization of SPS+ distilled images on CIFAR-100 with $M = 1$ stages. As privacy budget increases, distilled images have better visual fidelity.

Since our method produces data rather than a model, we include a second validation step in which a model is trained on the privatized dataset. By the DP post-processing property (Dwork, 2006), any choice of optimizer, model, or training configuration in this stage does not affect the privacy budget. This flexibility allows us to explore a range of options.

Figure 3: FID vs. Accuracy for CIFAR-10 and CIFAR=100 of SPS+ ($M = 2, 4$ for CF10/100, respectively)

Specifically, we fine-tune Wide ResNets of sizes 28-10 and 34-10 (both with ReLU activations), pretrained on $32\times32$ Downsampled ImageNet, using the GSAM optimizer. We also evaluate ensembles of $E = 5$ fine-tuned models. In contrast, standard DP-SGD pipelines impose much stricter limitations: ensembles would require additional composition, and larger models such as WRN-34-10 would incur extra privacy cost due to their higher parameter count.

Table 1 summarizes our results, alongside two key baselines: the state-of-the-art DP training algorithm from (De et al., 2022) and the best reported accuracy from DP generation, Private Evolution (Lin et al., 2024). **Comparisons to additional gradient- and generation-based methods are provided in section F**. For brevity, we present only the strongest gradient-based and generation-based baselines in the main text.

On CIFAR-10, SPS—especially with ensembling—consistently outperforms DP-SGD across all privacy budgets, but falls short on CIFAR-100. SPS+ matches or exceeds DP-SGD in every setting, particularly under strict privacy budgets with many classes (e.g., CIFAR-100 at $\epsilon = 1$). We further observe that images distilled by SPS using a WRN-22-8 transfer effectively to larger models never seen during distillation, demonstrating the flexibility of our approach. Unless otherwise noted, **we focus on SPS+ in all subsequent experiments**, due to its superior overall performance.

## 5.2 OUT OF DOMAIN IMAGE CLASSIFICATION

To validate SPS under public data domain mismatch, a common challenge in private learning, we tested on CAMELYON17 (Bandi), a histopathology dataset of lymph node sections annotated for metastatic cancer, using $64 \times 64$ ImageNet as pretraining data. We use SPS in this setting as in the binary classification case, the pseudo-class method does not apply. We generated 100k synthetic images (50k) for each class at resolution $64 \times 64$ (downsampled from $96 \times 96$), and

Table 2: CAMELYON17 Classification Performance ($\delta = 3 \cdot 10^{-6}$)

|  | Accuracy |
|---|---|
| Ours ($\epsilon = 8$) | **92.6%** |
| DP-Diffusion ($\epsilon = 10$) (Ghalebikesabi et al., 2023) | 91.1% |
| Private Evolution ($\epsilon = 7.56$) (Lin et al., 2024) | 79.6% |
| DP-SGD ($\epsilon = 10$) (Ghalebikesabi et al., 2023) | 90.5% |

evaluated our classification performance in table 2. We compare against existing baseline methods Private Evolution (Lin et al., 2024), DP-Diffusion (Ghalebikesabi et al., 2023) and DP-SGD. SPS successfully handles the setting where there is significant mismatch between the private and public datasets.

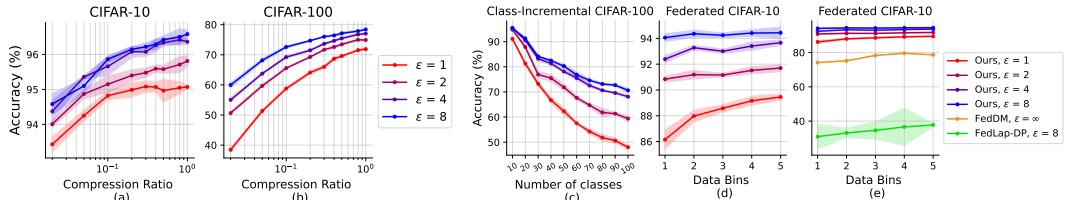

Figure 5: (a-b) Distillation performance of SPS+ at various compression ratios, trained on WRN34-10 models. High performance can be achieved with datasets of only 10% the size of the original, while remaining private. (c-e) capture Federated learning and Class-Incremental Continual learning performance of SPS+ on CIFAR-10 and CIFAR-100, respectively.

## 5.3 VISUALIZING THE DISTILLED IMAGES

A key advantage of data-based privacy is interpretability. Unlike DP training, which precludes data attribution methods (Koh & Liang, 2020), our method produces inspectable synthetic data. Fig 4 shows distilled CIFAR-100 images across privacy budgets: as $\epsilon$ increases, images evolve from abstract textures to recognizable class-specific representations, and consequently FID decreases. fig. 3 additionally visualizes the negative correlation between downstream performance and FID for SPS+.

## 5.4 SIMULTANEOUS DISTILLATION AND PRIVATIZATION

Table 3: Oversized synthesis performance of SPS+ on CIFAR-100. Oversized synthetic datasets can further improve performance

| | Distilled Dataset size | | | | |
|---|---|---|---|---|---|
| $\epsilon$ | 1× | 2× | 3× | 4× | DP-SGD |
| 1 | **76.6** | 76.4 | 76.0 | 75.9 | 70.3 |
| 2 | 79.2 | **79.4** | **79.4** | 79.3 | 74.7 |
| 4 | 80.7 | 81.2 | **81.3** | 81.1 | 79.2 |
| 8 | 81.6 | 81.8 | **82.1** | 81.9 | 81.8 |

In section 5.1, we evaluated performance when the distilled dataset had a $1 : 1$ compression ratio, i.e., equal in size to the original dataset. A notable property of our approach is that once the statistics $\tilde{v}$ are privatized, the number of synthetic images to generate can be chosen freely—without incurring any additional privacy cost. This enables simultaneous distillation and privatization by setting $|\mathcal{X}_S| < |\mathcal{X}_T|$. In this section, we investigate this setting using $M = 2$ for CIFAR-10 and $M = 4$ for CIFAR-100.

We follow the evaluation protocol of section 5.1, training a single WRN-34-10 model at various privacy budgets and reporting results in Fig 5. Larger synthetic datasets lead to higher accuracy, but even when using a privatized dataset with only 10% of the original size, performance on CIFAR-10 drops by merely $\sim 1\%$, highlighting the efficiency of our approach.

**Oversized dataset distillation**. SPS+ also allows synthetizing datasets *larger than the original dataset*. table 3 shows the effect of creating distilled datasets up to four times the size of the original, evaluated on WRN34-10 ensembles. For CIFAR-100, further performance gains is unlocked with oversized distilled datasets.

## 5.5 FEDERATED LEARNING

Federated learning enables multiple parties to collaboratively train models without sharing raw data. Traditional approaches exchange gradients, but these are vulnerable to reconstruction attacks (Zhu et al., 2019b) and require synchronized communication rounds. By contrast, our data-based approach enables *asynchronous* federated learning: each party independently generates privatized datasets using SPS+ and shares them without synchronization constraints.

We evaluate this by splitting CIFAR-10 into five partitions of 10,000 images each, with each party running SPS+ independently. Specifically each of $N = 5$ parties have a disjoint subset of the original dataset: $\mathcal{X}_T^1, ... \mathcal{X}_T^N$,, with $|\mathcal{X}_T^1| = 10,000$. Each party run SPS+ locally and independently, generating synthetic privatized sets $\mathcal{X}_S^1, ... \mathcal{X}_S^N$. These are sent to a server, which combines the datasets into a single one: $\mathcal{X}_S^{combined} = \bigcup_i^N \mathcal{X}_S^i$, and trains on $\mathcal{X}_S^{combined}$. SPS also supports a

variant that addresses this by performing *centralized generation* while still keeping only privatized statistics on the server, which we discuss in section B.6.

As shown in fig. 5, compared to FedLAP-DP (Wang et al., 2024) and FedDM (Xiong et al., 2022), Federated SPS+ significantly outperforms, particularly under strict privacy budgets (fig. 5). Federated SPS+ also successfully aggregates data from multiple sources, improving performance with more sources. For example, at $\epsilon = 1$, accuracy improves from 86% with a single data source to 89.5% with five sources.

### 5.6 CONTINUAL LEARNING

Continual learning involves training models sequentially on a stream of tasks or datasets. In the private setting, this presents a fundamental challenge: revisiting past data consumes additional privacy budget, while discarding it leads to catastrophic forgetting (French, 1999). Data-based privacy methods address this by enabling unlimited reuse of previously privatized datasets without incurring extra privacy cost.

We evaluate class-incremental learning on CIFAR-100 by splitting the dataset into 10 subsets and each has 10 classes. Each subset is privatized with SPS+ under budget $(\epsilon, \delta)$, and at stage $1 \leq \tau \leq 10$ we train on the cumulative privatized sets $\{\mathcal{X}_S^1, \ldots, \mathcal{X}_S^\tau\}$. Results show that performance remains close to the regular, non-continual training: for example, at $\epsilon = 4$, our method achieves $68.1 \pm 0.7\%$ accuracy, compared to $76.9 \pm 0.4\%$ for standard training (fig. 5).

## 6 LIMITATIONS, DISCUSSION, AND CONCLUSION

In this paper we present SPS and SPS+, algorithms which convert a dataset into a private version leveraging a model pretrained on public data. They are the dataset-distillation based which yields competitive results with fine-tuning based on DP-SGD. SPS also yields more flexibility than DP-SGD, at it supports tasks such as private federated learning and continual learning without modification.

Despite SPS's strong performance, there are several areas for improvement. The cost of generating these images is relatively heavy (see section F.1 for discussion). Future work could look at whether SPS generation can be amortized, or whether public generators could be used with SPS-style losses, similar to GLaD (Cazenavette et al., 2023) for dataset distillation. In this work we also focused on the simpler class-balanced setting, but future work could study SPS for classes with extreme class imbalance. Other work could look at extending SPS to discrete modalities such as text. Overall, SPS presents a promising new alternative for private deep learning which offers flexibility beyond DP-SGD.

## 7 REPRODUCIBILITY STATEMENT

Full algorithm pseudocode and descriptions are available in section A.1. Code is provided in the supplementary material.

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

## A  ALGORITHM DETAILS

### A.1  SPS ALGORITHM PSEUDOCODE

We break down the SPS algorithm into 3 stages: the *summarize* procedure, given in the `summarize` algorithm in algorithm 2, produces the (non-privatized) $\tilde{v}$ given a dataset; the privatization is done via by adding noise to $\tilde{v}$. Then, `process_v` converts $\tilde{v}$ into means and covariances $\mu_l$ and $\Sigma_l$ in algorithm 3. Finally, `synthesize` takes these statistics tp generates the synthetic images, described in algorithm 4. The master algorithm is given in algorithm 1. Note this method also uses data augmentation, similar to (Yin et al., 2023b) and (Loo et al., 2024), whose details are given in section D.

---

Algorithm 1: Summarize, Privatize, Synthesize (SPS)

**Input:** Dataset $\mathcal{X}_T$, and relevant parameters such as $N = |\mathcal{X}_T|$ and class count $C$
    Target distilled dataset size $S$
    Target privacy budget $(\epsilon, \delta)$
    Maximum $v$ clip ratio $K_{\text{clip}}$
    Pretrained model $\theta_P$
    Target privatized dataset labels for a single batch $\mathcal{Y}_B$, where $|\mathcal{Y}_B| = B$ and $B \mod C = 0$
**Output:** Privatized dataset $\mathcal{X}_S$
**BEGIN**
$b_0 \leftarrow \text{dp\_accountant}(\epsilon, \delta, \text{compositions} = 1)$ {calculate noise to add based on budget}
$|v|_{\max} \leftarrow K_{\text{clip}} \sqrt{2LD_G^{layer}}$ {Compute maximimum norm of $v$}
$\tilde{v}^T \leftarrow \text{summarize}(\mathcal{X}_T, \theta_P, K_{\text{clip}})$
$\tilde{v}^T \leftarrow \tilde{v}^T + \mathcal{N}(0, b_0^2 |v|_{\max}^2)$ {privatize with Gaussian Mechanism}
$\mathcal{X}_S = \{\}$
**while** $|\mathcal{X}_S| < S$ **do**
    $\mathcal{X}_S \leftarrow \mathcal{X}_S \cup \text{synthesize}(\tilde{v}^T, \theta_P, \mathcal{Y}_B)$ {Generate images until we have enough, one batch at a time}
**end while**
**RETURN** $\mathcal{X}_S$
**END**

---

Algorithm 2: `summarize`

**Input:** Dataset $\mathcal{X}_T$, labels $\mathcal{Y}_T$
    Maximum $v$ clip ratio $K_{\text{clip}}$
    Pretrained model $\theta_P$
    Random Projection Matrices $M_l^G$ for $l \in [L]$ with dimension $\mathbb{R}^{D_G \times D_l}$    Random Projection Matrices $M_l^C$ for $l \in L_C$ with dimension $\mathbb{R}^{D_C \times D_l}$
**Output:** Data global means and covariances
**BEGIN**
Compute $S \leftarrow \frac{LD_G^{layer}}{|L_C| D_C^{layer}}$
$|v|_{\max} \leftarrow K_{\text{clip}} \sqrt{2LD_G^{layer}}$
Init. $\tilde{v} = 0$
**for** $\{x_i, y_i\} \in \mathcal{X}_T, \mathcal{Y}_T$ **do**
    $z_{i,l} \leftarrow \text{intermediate\_reps}(\theta_P, x_i)$  $l \leq l \leq L$ {Compute pre-batchnorm representations of model}
    $z_{i,l}^G \leftarrow 2\sigma(M_l^G z_{i,l}) - 1 \in \mathbb{R}^{D_G \times H \times W}$  $l \leq l \leq L$
    $z_{i,l}^C \leftarrow 2\sigma(M_l^C z_{i,l}) - 1 \in \mathbb{R}^{D_C \times H \times W}$  $l \in L_C$ {Project to dimensions}
    $m_{i,l}^G \leftarrow \frac{1}{HW} \sum_{h=1}^H \sum_{w=1}^W z_{i,l}^G \in R^{D_G}$,  $1 \leq l \leq L$
    $s_{i,l}^G \leftarrow \frac{1}{HW} \sum_{h=1}^H \sum_{w=1}^W z_{i,l}^G z_{i,l}^{\intercal G} \in \mathbb{R}^{D_G \times D_G}$,  $1 \leq l \leq L$
    $m_{i,l}^c \leftarrow \frac{1_{(y_i=c)}}{HW} \sum_{h=1}^H \sum_{w=1}^W z_{i,l}^C \in \mathbb{R}^{D_C}$,  $l \in L_C, 1 \leq c \leq C$
    $s_{i,l}^c \leftarrow \frac{1_{(y_i=c)}}{HW} \sum_{h=1}^H \sum_{w=1}^W z_{i,l}^C z_{i,l}^{\intercal C} \in \mathbb{R}^{D_C \times D_C}$,  $l \in L_C, 1 \leq c \leq C$ {eq. (3)}
    $u_{i,l}^G \leftarrow \text{triu}(s_{i,l}^G)$,  $1 \leq l \leq L$
    $u_{i,l}^c \leftarrow \text{triu}(s_{i,l}^c)$,  $l \in L_C, 1 \leq c \leq C$ {Get Upper Triangular Part}
    $m_{i,l}^c \leftarrow \sqrt{S} m_{i,l}^c, u_{i,l}^c \leftarrow \sqrt{S} u_{i,l}^c$,  $l \in L_C, 1 \leq c \leq C$ {Scale per-class statistics}
    Set $v_i$ as eq. (3)
    $v_i \leftarrow \text{clip}_{|v|_{\max}}(v_i)$ {Clip with eq. (4)}
    $\tilde{v} \leftarrow \tilde{v} + v_i$ {Add to $\tilde{v}$}
**end for**
**RETURN** $\tilde{v}$
**END**

---

---

**Algorithm 3:** `process_v`, Converts $\tilde{v}$ into raw means and covariances

---

**Input:** Summary vector $\tilde{v}$
Dataset count $N$ and class count $C$
Jitter parameter $\gamma_{\text{eig}}$
**Output:** Summary vector $\tilde{v}$
**BEGIN**
Compute $S \leftarrow \frac{LD_G^{layer}}{|L_C|D_C^{layer}}$
Split $\tilde{v}$ according to eq. (3), $\tilde{m}_l^G, \tilde{u}_l^G$ for each layer and $\tilde{m}_l^c, \tilde{u}_l^c$ for each class and layer.
**for** $l \in [L]$ **do**
$\quad \mu_l^G \leftarrow \frac{1}{N}\tilde{m}_l^G$
$\quad \hat{\Sigma}_l^G \leftarrow \frac{1}{N}\text{triu}^{-1}(\tilde{u}_l^G) - \mu_l^G \mu_l^{\intercal G}$
$\quad$ **if** Noise added to $\tilde{v}$ **then**
$\quad\quad \sigma_l^G = -\gamma_{\text{eig}} \cdot \min(\lambda(\hat{\Sigma}_l^G))$ {Choose jitter parameter based on minimum eigenvalue of $\hat{\Sigma}$}
$\quad\quad \Sigma_l^G \leftarrow \text{eigclip}(\hat{\Sigma}_l^G) + \sigma_l^G I$
$\quad$ **else**
$\quad\quad \Sigma_l^G \leftarrow \hat{\Sigma}_l^G$ {If no noise is added, then $\Sigma_l^G$ will be positive definite}
$\quad$ **end if**
**end for**
**for** $l \in L_C, c \in [C]$ **do**
$\quad \mu_l^c \leftarrow \frac{C}{N\sqrt{S}}\tilde{m}_l^c$ {Additionally unscale by $\sqrt{S}$}
$\quad \hat{\Sigma}_l^c \leftarrow \frac{C}{N\sqrt{S}}\text{triu}^{-1}(\tilde{u}_l^c) - \mu_l^c\mu_l^{\intercal G}$
$\quad$ **if** Noise added to $\tilde{v}$ **then**
$\quad\quad \sigma_l^c = -\gamma_{\text{eig}} \cdot \min(\lambda(\hat{\Sigma}_l^c))$ {Choose jitter parameter based on minimum eigenvalue of $\hat{\Sigma}$}
$\quad\quad \Sigma_l^c \leftarrow \text{eigclip}(\hat{\Sigma}_l^c) + \sigma_l^c I$
$\quad$ **else**
$\quad\quad \Sigma_l^c \leftarrow \hat{\Sigma}_l^c$ {If no noise is added, then $\Sigma_l^c$ will be positive definite}
$\quad$ **end if**
**end for**
**RETURN** $\{\mu_l^G, \Sigma_l^G\}_{l\in[L]}, \{\mu_l^c, \Sigma_l^c\}_{l\in L_C, c\in[C]}, \{\sigma_l^G\}_{l\in[L]}, \{\sigma_l^c\}_{l\in L_C, c\in[C]}$
**END**

---

---

**Algorithm 4:** `synthesize`, Converts summary statistics $\tilde{v}$ into a batch of synthetic images

---

**Input:** $M$ summary vectors and models $\{\tilde{v}^{T,m}\}_{m=1}^M, \{\theta_P^m\}_{m=1}^M$
Max iterations per batch $I$
Batch target labels $\mathcal{Y}_B$
Learning rate $\eta$
Class kl-divergence coefficient $\lambda_C$
Data Augmentation function `aug`
Maximum $v$ clip ratio $K_{\text{clip}}$
**Output:** Batch of privatized images $\mathcal{X}_B$
**BEGIN**
$\{\mu_{l,m}^{T,G}, \Sigma_{l,m}^{T,G}\}_{l\in[L]}, \quad \{\mu_{l,m}^{T,c}, \Sigma_{l,m}^{T,c}\}_{l\in L_C, c\in[C]}, \quad \{\sigma_{l,m}^{T,G}\}_{l\in[L]}, \quad \{\sigma_{l,m}^{T,c}\}_{l\in L_C, c\in[C]} =$ `process_v`$(\tilde{v}^{T,m})$ for $m \in [M]$
{Get means and covariances from summary stats}
$\mathcal{X}_B \leftarrow \mathcal{N}(0, I)$ {Init distilled dataset as random noise}
**for** $i \in [I]$ {Optimize $I$ steps} **do**
$\quad m \leftarrow i \mod M$ {Select a model}
$\quad \tilde{v}^{B,m} \leftarrow$ `summarize`$(\text{aug}(\mathcal{X}_B), \theta_P^m, K_{\text{clip}})$ {Apply data augmentation and compute summary stats}
$\quad \{\mu_{l,m}^{G,B}, \Sigma_{l,m}^{G,B}\}_{l\in[L]}, \{\mu_{l,m}^{c,B}, \Sigma_{l,m}^{c,B}\}_{l\in L_C, c\in[C]} =$ `process_v`$(\tilde{v}^{B,m})$
$\quad L_{\text{SPS}} \leftarrow \sum_{l=1}^L D_{KL}\Big(\mathcal{N}(\mu_l^{T,G}, \Sigma_l^{T,G})||\mathcal{N}(\mu_l^{B,G}, \Sigma_l^{B,G} + \sigma_{l,m}^{T,G}I)\Big)$
$\quad\quad + \lambda_C \sum_{c=1}^C \sum_{l\in L_C} D_{KL}\Big(\mathcal{N}(\mu_l^{T,c}, \Sigma_l^{T,c})||\mathcal{N}(\mu_l^{B,c}, \Sigma_l^{B,c} + \sigma_{l,m}^{T,c}I)\Big)$
$\quad$ {eq. (2), with jitters applied to batch covariances}
$\quad \mathcal{X}_B \leftarrow \mathcal{X}_B - \eta \frac{\partial \mathcal{L}_{\text{SPS}}}{\partial \mathcal{X}_B}$
**end for**
**RETURN** $\mathcal{X}_B$
**END**

---

## A.2 SPS+ ALGORITHM PSEUDOCODE

SPS+ proceeds similarly to SPS, but additionally adds an outer stage loop. See algorithm 5 for details. We use modified versions of the SPS subroutines given in algorithm 5 -algorithm 7. Changes to the original pseudocode are highlighted in red.

---

Algorithm 5: Summarize, Privatize, Synthesize Plus (SPS+)

---

**Input:** Dataset $\mathcal{X}_T$, and relevant parameters such as $N = |\mathcal{X}_T|$ and class count $C$
  Target distilled dataset size $S$
  Target privacy budget $(\epsilon, \delta)$
  Maximum $v$ clip ratios $\{K_{\text{clip}}^m\}_{m=1}^M$
  $M$ pretrained models $\{\theta_P^m\}_{m=1}^M$
  Target number of stages, $M$
  Target privatized dataset labels for a single batch $\mathcal{Y}_B$, where $|\mathcal{Y}_B| = B$ and $B \mod C = 0$
**Output:** Privatized dataset $\mathcal{X}_S$
**BEGIN**
$b_0 \leftarrow$ dp_accountant$(\epsilon, \delta, \text{compositions} = M)$ {calculate noise to add based on budget}

$|v|_{\max}^m \leftarrow K_{\text{clip}}^m \sqrt{2LD_G^{layer}}$ {Compute maximimum norm of $v$}
$\mathcal{V}_{\text{prior}} \leftarrow \{0\}$ {Initialize set of prior stats with zero for first round}
$\mathcal{X}_S \leftarrow \mathcal{N}(0, I)$ {Initialize images as random noise}
**for** $m_{\text{outer}} = 1$ **to** $M$ **do**
  **if** $m_{\text{outer}} > 1$ **then**
    $\tilde{v}_{\text{prior}}^{m_{\text{outer}}} \leftarrow$ summarize_plus$(\mathcal{X}_S, \theta_P^{m_{\text{outer}}}, \infty, 0)$ {Get unnormalized unclipped summary statistic from existing synthetic images}
    $v_{\text{prior}}^{m_{\text{outer}}} \leftarrow$ normalize$(\tilde{v}_{\text{prior}}^{m_{\text{outer}}})$ {normalize with eq. (5)}
    $\mathcal{V}_{\text{prior}} \leftarrow \mathcal{V}_{\text{prior}} \cup \{v_{\text{prior}}^{m_{\text{outer}}}\}$ {Add prior summary statistics of current synthetic images to set of prior stats}
  **end if**
  $\mathcal{X}_S^{\text{inner}} \leftarrow \{\}$
  $\tilde{v}^{T, m_{\text{outer}}} \leftarrow$ summarize_plus$(\mathcal{X}_T, \theta_P^{m_{\text{outer}}}, K_{\text{clip}}^{m_{\text{outer}}}, \mathcal{V}_{\text{prior}}^m{}_{\text{outer}})$
  $\tilde{v}^{T, m} \leftarrow \tilde{v}^{T, m} + \mathcal{N}(0, b_0^2 |v|_{\max}^2)$ {privatize with Gaussian Mechanism}
  **while** $|\mathcal{X}_S| < S$ **do**
    $\mathcal{X}_B^{\text{init}} \leftarrow$ sample$(\mathcal{X}_S)$ {select initialization images for batch}
    $\mathcal{X}_S^{\text{inner}} \leftarrow \mathcal{X}_S^{\text{inner}} \cup$ synthesize_plus$(\{\tilde{v}^{T,m}\}_{m=1}^{m_{\text{outer}}}, \{\theta_P^m\}_{m=1}^{m_{\text{outer}}}, \mathcal{Y}_B, \mathcal{V}_{\text{prior}}, \mathcal{X}_B^{\text{init}})$ {Generate images until we have enough, one batch at a time}
  **end while**
  $\mathcal{X}_S \leftarrow \mathcal{X}_S^{\text{inner}}$
**end for**
**RETURN** $\mathcal{X}_S$
**END**

---

---

Algorithm 6: summarize_plus (for SPS+)

---

**Input:** Dataset $\mathcal{X}_T$, labels $\mathcal{Y}_T$
  Maximum $v$ clip ratio $K_{\text{clip}}$
  Pretrained model $\theta_P$
  Random Projection Matrices $M_l^G$ for $l \in [L]$ with dimension $\mathbb{R}^{D_G \times D_l}$
  Random Projection Matrices $M_l^C$ for $l \in L_C$ with dimension $\mathbb{R}^{D_C \times D_l}$
  Psuedoclass count P
  Pseudoclass matrix $\boldsymbol{P} \in \mathbb{R}^{C \times P}$
  Prior statistics $v_{\text{prior}}$
**Output:** Data global means and covariances
**BEGIN**
Split $v_{\text{prior}}$ according to eq. (3), $m_l^{G_{\text{prior}}}, u_l^{G_{\text{prior}}}$ for each layer and $m_l^{P_{\text{prior}}}, u_l^{P_{\text{prior}}}$ for each pseudo-class and layer. (note these quantities are normalized)
Compute $S \leftarrow \frac{LD_G^{layer}}{|L_C|D_C^{layer}}$

$|v|_{\max} \leftarrow K_{\text{clip}} \sqrt{2LD_G^{layer}}$
Init. $\tilde{v} = 0$
**for** $\{x_i, y_i\} \in \mathcal{X}_T, \mathcal{Y}_T$ **do**
  $z_{i,l} \leftarrow$ intermediate_reps$(\theta_P, x_i)$   $l \le l \le L$ {Compute pre-batchnorm representations of model}
  $z_{i,l}^G \leftarrow 2\sigma(M_l^G z_{i,l}) - 1 \in \mathbb{R}^{D_G \times H \times W}$   $l \le l \le L$
  $z_{i,l}^C \leftarrow 2\sigma(M_l^C z_{i,l}) - 1 \in \mathbb{R}^{D_C \times H \times W}$   $l \in L_C$ {Project to dimensions}
  $m_{i,l}^G \leftarrow \frac{1}{HW} \sum_{h=1}^H \sum_{w=1}^W z_{i,l}^G - m_l^{G_{\text{prior}}} \in R^{D_G}$,   $1 \le l \le L$
  $s_{i,l}^G \leftarrow \frac{1}{HW} \sum_{h=1}^H \sum_{w=1}^W z_{i,l}^G z_{i,l}^{\mathsf{T}G} - \text{triu}^{-1}(u_l^{G_{\text{prior}}}) \in \mathbb{R}^{D_G \times D_G}$,   $1 \le l \le L$
  $m_{i,l}^p \leftarrow \mathbb{1}_{\boldsymbol{P}_{y_i,p}=1} \left( \frac{1}{HW} \sum_{h=1}^H \sum_{w=1}^W z_{i,l}^c - m_l^{P_{\text{prior}}} \right) \in \mathbb{R}^{D_C}$,   $l \in L_C, 1 \le p \le P$
  $s_{i,l}^p \leftarrow \mathbb{1}_{\boldsymbol{P}_{y_i,p}=1} \left( \frac{1}{HW} \sum_{h=1}^H \sum_{w=1}^W z_{i,l}^C z_{i,l}^{\mathsf{T}C} - \text{triu}^{-1}(u_l^{P_{\text{prior}}}) \right) \in \mathbb{R}^{D_C \times D_C}$,   $l \in L_C, 1 \le p \le P$ {eq. (3)}
  $u_{i,l}^G \leftarrow \text{triu}(s_{i,l}^G)$,   $1 \le l \le L$
  $u_{i,l}^c \leftarrow \text{triu}(s_{i,l}^c)$,   $l \in L_C, 1 \le c \le C$ {Get Upper Triangular Part}
  $m_{i,l}^c \leftarrow \sqrt{S} m_{i,l}^c, u_{i,l}^c \leftarrow \sqrt{S} u_{i,l}^c$,   $l \in L_C, 1 \le c \le C$ {Scale per-class statistics}
  Set $v_i$ as eq. (3)
  $v_i \leftarrow \text{clip}_{|v|_{\max}}(v_i)$ {Clip with eq. (4)}
  $\tilde{v} \leftarrow \tilde{v} + v_i$ {Add to $\tilde{v}$}
**end for**
**RETURN** $\tilde{v}$
**END**

---

---

Algorithm 7: `process_v_plus`, Converts $\tilde{v}$ into raw means and covariances

---

**Input:** Summary vector $\tilde{v}$
Dataset count $N$ and class count $C$
Jitter parameter $\gamma_{\text{eig}}$
Normalized prior statistics $v_{\text{prior}}$
**Output:** Summary means and covariances $\{\mu_l^G, \Sigma_l^G\}_{l \in [L]}$, $\{\mu_l^p, \Sigma_l^p\}_{l \in L_C, p \in [P]}$, $\{\sigma_l^G\}_{l \in [L]}$, $\{\sigma_l^p\}_{l \in L_C, p \in [P]}$

**BEGIN**
Compute $S \leftarrow \frac{L D_G^{layer}}{|L_C| D_C^{layer}}$
Split $v_{\text{prior}}$ according to eq. (3), $m_l^{G_{\text{prior}}}, u_l^{G_{\text{prior}}}$ for each layer and $m_l^{p_{\text{prior}}}, u_l^{p_{\text{prior}}}$ for each pseudo-class and layer. (note these quantities are normalized)
Split $\tilde{v}$ according to eq. (3), $\tilde{m}_l^G, \tilde{u}_l^G$ for each layer and $\tilde{m}_l^p, \tilde{u}_l^p$ for each pseudo-class and layer.
**for** $l \in [L]$ **do**
$\quad \mu_l^G \leftarrow \frac{1}{N} \tilde{m}_l^G + m_l^{G_{\text{prior}}}$
$\quad \hat{\Sigma}_l^G \leftarrow \frac{1}{N} \text{triu}^{-1}(\tilde{u}_l^G) - \mu_l^G \mu_l^{\mathsf{T}G} + \text{triu}^{-1}(u_l^{G_{\text{prior}}})$
$\quad$ **if** Noise added to $\tilde{v}$ **then**
$\quad\quad \sigma_l^G = -\gamma_{\text{eig}} \cdot \min(\lambda(\hat{\Sigma}_l^G))$ {Choose jitter parameter based on minimum eigenvalue of $\hat{\Sigma}$}
$\quad\quad \Sigma_l^G \leftarrow \text{eigclip}(\hat{\Sigma}_l^G) + \sigma_l^G I$
$\quad$ **else**
$\quad\quad \Sigma_l^G \leftarrow \hat{\Sigma}_l^G$ {If no noise is added, then $\Sigma_l^G$ will be positive definite}
$\quad$ **end if**
**end for**
**for** $l \in L_C, p \in [P]$ **do**
$\quad \mu_l^p \leftarrow \frac{C}{N N_{C/p} \sqrt{S}} \tilde{m}_l^p + m_l^{p_{\text{prior}}}$ {Additionally unscale by $\sqrt{S}$}
$\quad \hat{\Sigma}_l^p \leftarrow \frac{C}{N N_{C/p} \sqrt{S}} \text{triu}^{-1}(\tilde{u}_l^c) - \mu_l^p \mu_l^{\mathsf{T}p} + \text{triu}^{-1}(u_l^{p_{\text{prior}}})$
$\quad$ **if** Noise added to $\tilde{v}$ **then**
$\quad\quad \sigma_l^p = -\gamma_{\text{eig}} \cdot \min(\lambda(\hat{\Sigma}_l^p))$ {Choose jitter parameter based on minimum eigenvalue of $\hat{\Sigma}$}
$\quad\quad \Sigma_l^p \leftarrow \text{eigclip}(\hat{\Sigma}_l^p) + \sigma_l^p I$
$\quad$ **else**
$\quad\quad \Sigma_l^p \leftarrow \hat{\Sigma}_l^c$ {If no noise is added, then $\Sigma_l^p$ will be positive definite}
$\quad$ **end if**
**end for**
**RETURN** $\{\mu_l^G, \Sigma_l^G\}_{l \in [L]}, \{\mu_l^p, \Sigma_l^p\}_{l \in L_C, p \in [P]}, \{\sigma_l^G\}_{l \in [L]}, \{\sigma_l^p\}_{l \in L_C, p \in [P]}$
**END**

---

---

Algorithm 8: `synthesize`, Converts summary statistics $\tilde{v}$ into a batch of synthetic images

---

**Input:** $M$ summary vectors and models $\{\tilde{v}^{T,m}\}_{m=1}^M, \{\theta_P^m\}_{m=1}^M$
Max iterations per batch $I$
Batch target labels $\mathcal{Y}_B$
Learning rate $\eta$
Class kl-divergence coefficient $\lambda_C$
Data Augmentation function `aug`
Maximum $v$ clip ratios $\{K_{\text{clip}}^m\}_{m=1}^M$
$M$ prior summary vectors and models $\{\tilde{v}_{\text{prior}}^{T,m}\}_{m=1}^M$
Init images $\mathcal{X}_B^{\text{init}}$
**Output:** Batch of privatized images $\mathcal{X}_B$
**BEGIN**
$\{\mu_{l,m}^{T,G}, \Sigma_{l,m}^{T,G}\}_{l \in [L]}, \{\mu_{l,m}^{T,c}, \Sigma_{l,m}^{T,c}\}_{l \in L_C, p \in [P]}, \{\sigma_{l,m}^{T,G}\}_{l \in [L]}, \{\sigma_{l,m}^{T,c}\}_{l \in L_C, p \in [P]}$
$\quad \leftarrow \text{process\_v\_plus}(^{T,m}, \tilde{v}_{\text{prior}}^{T,m})$ for $m \in [M]$
{Get means and covariances from summary stats}
$\mathcal{X}_B \leftarrow \mathcal{X}_B^{\text{init}}$ {Init distilled dataset}
**for** $i \in [I]$ {Optimize $I$ steps} **do**
$\quad m \leftarrow i \mod M$ {Select a model}
$\quad \tilde{v}^{B,m} \leftarrow \text{summarize\_plus}(\text{aug}(\mathcal{X}_B), \theta_P^m, K_{\text{clip}}^m, \tilde{v}_{\text{prior}}^{T,m})$ {Apply data augmentation and compute summary stats}
$\quad \{\mu_{l,m}^{G,B}, \Sigma_{l,m}^{G,B}\}_{l \in [L]}, \{\mu_{l,m}^{p,B}, \Sigma_{l,m}^{p,B}\}_{l \in L_C, p \in [P]} = \text{process\_v\_plus}(\tilde{v}^{B,m}, \tilde{v}_{\text{prior}})$
$\quad L_{\text{SPS}} \leftarrow \sum_{l=1}^L D_{KL}\Big(\mathcal{N}(\mu_l^{T,G}, \Sigma_l^{T,G}) || \mathcal{N}(\mu_l^{B,G}, \Sigma_l^{B,G} + \sigma_{l,m}^{T,G} I)\Big)$
$\quad\quad\quad + \lambda_C \sum_{p=1}^P \sum_{l \in L_C} D_{KL}\Big(\mathcal{N}(\mu_l^{T,p}, \Sigma_l^{T,p}) || \mathcal{N}(\mu_l^{B,c}, \Sigma_l^{B,p} + \sigma_{l,m}^{T,p} I)\Big)$
$\quad$ {eq. (2), with jitters applied to batch covariances}
$\quad \mathcal{X}_B \leftarrow \mathcal{X}_B - \eta \frac{\partial \mathcal{L}_{\text{SPS}}}{\partial \mathcal{X}_B}$
**end for**
**RETURN** $\mathcal{X}_B$
**END**

---

## A.3 CLIPPING EIGENVALUES

As discussed in section 3, and used in algorithm 3, we clip eigenvalues of $\hat{\Sigma}$. This is necessary to be a valid covariance matrix, as covariance matrices must be positive definite. This is done by performing an eigendecomposition of $\hat{\Sigma}$ and setting negative eigenvalues to 0. This is only required for the noisy full-dataset statistics, since the noise can cause the estimated covariance to

have negative eigenvalues. This is not required to be done at each iteration step in algorithm 4, since we do not add noise to the synthetic images statistics. We also add a positive constant along the diagonal of size $\sigma$, whose magnitude is determine by the largest negative eigenvalue of $\hat{\Sigma}$, in a form $\Sigma = \text{eigclip}(\hat{\Sigma}) + \sigma I$, where eigclip clips all negative eigenvalues to 0.

## A.4 PROJECTION MATRIX CONSTRUCTION

As used in algorithm 2, we require random projection matrices $M_l^G \in \mathbb{R}^{D_G \times D_l}$ and $M_l^C \in \mathbb{R}^{D_C \times D_l}$. These are computed once and used through the algorithm (but are different for each model $m$). We sample entries of $M_l^G$ i.i.d. from $\mathcal{N}(0, \frac{1}{D_l})$, and likewise for $M_l^C$, to preserve the magnitude of the activations pre and post projection.

## A.5 GROUPED PSEUDO-CLASSES

In this section we describe the grouped pseudo-class method. Let use define a pseudo-class assignment matrix $\boldsymbol{P} \in \mathbb{R}^{C \times P}$ to be a binary matrix, where $\mathbb{1}_C^\intercal \boldsymbol{P} = N_{C/P}$, i.e. there are $N_{C/P}$ classes per pseudo-class. We generate these matrices randomly so that pseudo-classes are randomly assigned. Also assume that the pseudo-classes per class are the same, so that the pseudo-class assignments are balanced. One such example of the identity matrix $\boldsymbol{I}_C$, which trivially corresponds to using the original class assignments. As we can see from eq. (7) and algorithm 2, statistics for the pseudo-classes are done by normalizing the sum by $\frac{C}{N_{C/p}}$. As the sum is subject to a constant noise rate $b_0$, the relatively noise of the pseudo-class mean will decrease is $N_{C/p}$ increases. We also note that provided that $\boldsymbol{P}$ is invertible, then we can recover estimates of the original class means. Optimization proceeds by matching the pseudo-class statistics on the synthetic dataset $\mathcal{X}_S$ to those of the original private data $\mathcal{X}_T$.

We note that from a privacy perspective, the noise on the original class statistics does not improve by using pseudoclasses, and inverted by $\boldsymbol{P}^{-1}$. However, empirically, as seen in table 8, using pseudoclasses greatly improves the optimization dynamics of our loss function.

## A.6 MULTISTAGE CLIPPING

Algorithm 5 shows the modified algorithm with multistage clipping. The principle is heavily inspired by Biswas et al. (2022); Bie et al. (2023) The working intuition is that clipping around 0 should only be done if we have no prior guess of what $\tilde{v}$ should be around approximately. If we know what $\tilde{v}$ is approximately, we can center the clipping radius around there, and using a clipping radius. Multistage clipping proceeds as follows: firstly, we generate an initialize set of images with SPS, $\mathcal{X}_S^1$. Next, we measure global and per-class statistics of $\mathcal{X}_S^1$ as $\tilde{v}_{\text{prior}}^1$, with **no clipping and no noise**. We then need to normalize this, going from $\tilde{v}_{\text{prior}}^1$ to $v_{\text{prior}}^1$ as follows:

$$
\tilde{v}_{\text{prior}} = [\underbrace{\bar{m}_{1,\text{prior}}^G, \bar{u}_{1,\text{prior}}^G, \ldots, \bar{m}_{L,\text{prior}}^G, \bar{u}_{L,\text{prior}}^G}_{\tilde{v}_{\text{prior}}^G}, \underbrace{\bar{m}_{1,\text{prior}}^1, \bar{u}_{1,\text{prior}}^1, \ldots, \bar{m}_{L,\text{prior}}^1, \bar{u}_{L,\text{prior}}^1}_{\tilde{v}_{\text{prior}}^{p=1}} \ldots \underbrace{\bar{m}_{1,\text{prior}}^P, \bar{u}_{1,\text{prior}}^P, \ldots, \bar{m}_{L,\text{prior}}^P, \bar{u}_{L,\text{prior}}^P}_{\tilde{v}_{\text{prior}}^{p=P}}]
$$

$$
v_{\text{prior}} = [\tfrac{1}{N}\tilde{v}_{\text{prior}}^G, \tfrac{C}{N N_{C/p}\sqrt{S}}\tilde{v}_{\text{prior}}^{p=1}, \ldots, \tfrac{C}{N N_{C/p}\sqrt{S}}\tilde{v}_{\text{prior}}^{p=P}] \tag{5}
$$

$$
v_{\text{prior}} = [\underbrace{m_{1,\text{prior}}^G, u_{1,\text{prior}}^G, \ldots, m_{L,\text{prior}}^G, u_{L,\text{prior}}^G}_{v_{\text{prior}}^G}, \underbrace{m_{1,\text{prior}}^1, u_{1,\text{prior}}^1, \ldots, m_{L,\text{prior}}^1, u_{L,\text{prior}}^1}_{v_{\text{prior}}^{p=1}} \ldots \underbrace{m_{1,\text{prior}}^P, u_{1,\text{prior}}^P, \ldots, m_{L,\text{prior}}^P, u_{L,\text{prior}}^P}_{v_{\text{prior}}^{p=P}}]
$$

Note this unclipped measurement can be done without privacy loss, as we are only using $\mathcal{X}_S^1$. Concretely, the $m_l^G, u_l^G, m_l^p, u_l^p$ from $\mathcal{X}_S^1$, serves as approximate values for the true $m_l^G, u_l^G, m_l^c, u_l^c$ from $\mathcal{X}_T$, our private dataset. Like to Biswas et al. (2022); Bie et al. (2023), we use this as our new clipping center for the next round measurement. Specifically, we use the following update rules for $m_l^G, s_l^G, m_l^c, s_l^c$:

$$m_{i,l}^G = \frac{1}{HW} \sum_{h=1}^{H} \sum_{w=1}^{W} z_{i,l}^G - m_l^{G_{\text{prior}}} \in R^{D_G}, \quad 1 \le l \le L$$

$$s_{i,l}^G = \frac{1}{HW} \sum_{h=1}^{H} \sum_{w=1}^{W} z_{i,l}^G z_{i,l}^{\intercal G} - \text{triu}^{-1}(u_l^{G_{\text{prior}}}) \in \mathbb{R}^{D_G \times D_G}, \quad 1 \le l \le L$$

$$m_{i,l}^p = \mathbb{1}_{\boldsymbol{P}_{y_i,p}=1} \left( \frac{1}{HW} \sum_{h=1}^{H} \sum_{w=1}^{W} z_{i,l}^c - m_l^{p_{\text{prior}}} \right) \in \mathbb{R}^{D_C}, \quad l \in L_C, 1 \le p \le P$$

$$s_{i,l}^p = \mathbb{1}_{\boldsymbol{P}_{y_i,p}=1} \left( \frac{1}{HW} \sum_{h=1}^{H} \sum_{w=1}^{W} z_{i,l}^C z_{i,l}^{\intercal C} - \text{triu}^{-1}(u_l^{p_{\text{prior}}}) \right) \in \mathbb{R}^{D_C \times D_C}, \quad l \in L_C, 1 \le p \le P \quad (6)$$

Changes from standard SPS are highlighted in red. And later, after clipping, noising, and computing the mean, we readd the prior statistics.

$$\mu_l^p = \frac{C}{N N_{C/p} \sqrt{S}} \tilde{m}_l^p + m_l^{p_{\text{prior}}}$$

$$\hat{\Sigma}_l^p = \frac{C}{N N_{C/p} \sqrt{S}} \text{triu}^{-1}(\tilde{u}_l^c) - \mu_l^p \mu_l^{\intercal p} + \text{triu}^{-1}(u_l^{p_{\text{prior}}}) \quad (7)$$

# B    ABLATIONS

## B.1    CHOICE OF CLASS-MATCHING LOSS

In section 3, we mentioned that D3S heavily relies of soft labels to improve performance. We chose to use per-class distribution matching instead of cross-entropy loss to specifically address the limitations of hard labels in contrast to the soft teacher labels used by D3S. To evaluate the effect of this change, we performed additional ablations comparing our modifications to the D3S framework. D3S depends on soft labels produced by a trained teacher model, which cannot be generated privately. Our teacher models achieved accuracies of 98.1% on CIFAR-10 and 86.1% on CIFAR-100. All models were evaluated using a Wide ResNet-34-10 architecture, and were generated using five Wide ResNet-22-8 models. For D3S, these WRN-22-8 models were also trained on the source dataset.

Table 4 shows that the SPS loss—which matches per-class distributions rather than minimizing cross-entropy—significantly outperforms the D3S loss under hard labels. This is likely because SPS encourages distributional alignment across classes, instead of simply pushing embeddings toward the predicted logit. However, when soft labels are available, both approaches perform similarly. Thus, the use of per-class KL divergence in SPS is a deliberate and non-trivial adaptation to better handle the hard-label regime.

## B.2    CHOICE OF DATASET DISTILLATION ALGORITHM

As discussed in section 2.3, we discuss why we chose D3S over other dataset distillation algorithms. Here we discuss this more in detail. D3S offers two key advantages:

1. **Does not require training trajectories.** One of the most widely used dataset distillation approaches is Matching Training Trajectories (MTT)(Cazenavette et al., 2022), along with several of its variants (Li et al., 2025; Liu et al., 2024). These methods require access to full training trajectories from a model trained on the private dataset, which would need to be obtained using DP-SGD. Since DP-SGD would be required as a subroutine, these methods are impractical for our goal of improving over DP-SGD in either performance or efficiency.

2. **Allows unlimited optimization steps per fixed number of measurements.** Many distillation algorithms interact with the private dataset at every optimization step—e.g., by

Table 4: Performance of the SPS algorithms with changes ablated, evaluated on CIFAR-10 and CIFAR-100 on WRN34-10 models. We see that class rescaling, the use of SiLU activations, and using the GSAM optimizer at validation are necessary for high performance.

|  | CIFAR-10 | CIFAR-100 |
|---|---|---|
| D3S (with soft labels, $\epsilon = \infty$) | 97.7 | 84.9 |
| SPS (with soft labels, $\epsilon = \infty$) | 97.7 | 84.5 |
| D3S (hard labels, $\epsilon = \infty$) | 94.6 | 65.8 |
| SPS (hard labels, $\epsilon = \infty$) | 96.6 | 76.9 |
| SPS (hard labels, $\epsilon = 8$) | 96.1 | 72.2 |

Table 5: Performance of the SPS algorithms with optimization changes ablated, evaluated on CIFAR-10 on WRN28-10 models.

| Modification | CIFAR-10 | | | |
|---|---|---|---|---|
|  | $\epsilon = 1$ | $\epsilon = 2$ | $\epsilon = 4$ | $\epsilon = 8$ |
| SPS | $\mathbf{93.2 \pm 0.2}$ | $\mathbf{94.6 \pm 0.2}$ | $\mathbf{95.0 \pm 0.2}$ | $\mathbf{95.9 \pm 0.1}$ |
| ✗ Class Rescaling | $89.1 \pm 0.3$ | $92.4 \pm 0.4$ | $88.8 \pm 0.8$ | $93.1 \pm 0.4$ |
| ✗ Smooth Activation | $90.7 \pm 0.3$ | $93.6 \pm 0.4$ | $92.1 \pm 0.8$ | $95.4 \pm 0.4$ |
| ✗ GSAM | $89.9 \pm 0.8$ | $92.6 \pm 0.2$ | $92.8 \pm 0.5$ | $95.0 \pm 0.2$ |

computing embeddings from the private data—which consumes privacy budget with each access and leads to large composition penalties under DP. However, dataset distillation typically requires thousands of optimization steps, making these approaches inefficient or even infeasible under privacy constraints. This limits us to "decoupled" distillation methods such as D3S, or SRe2L (Loo et al., 2024; Yin et al., 2023a), which decouple synthetic image optimization from direct interaction with the private dataset.

In practice, SRe2L performs poorly without access to teacher pseudo-labels. For example, under $\epsilon = \infty$, replacing the SPS loss with the SRe2L batchnorm-based loss results in CIFAR-10 accuracy of only 89.8%, which is lower than that of SPS at $\epsilon = 1$.

Moreover, there have been efforts to privatize non-decoupled methods, such as DP-KIP(Vinaroz & Park, 2024), which is based on (Nguyen et al., 2021a), but these perform poorly (see section F). Similarly, FedLAP-DP (based on DC (Zhao et al., 2021)) and FED-DM (based on (Zhao & Bilen, 2022)) perform significantly worse than SPS (see fig. 5).

### B.3 SPS Optimizaton Ablations

In section 3, we proposed a three additional changes to the algorithm to improve it: the use of class-rescaling to redistribute the noise, the use of smooth SiLU activations in the pretrained model to improve optimization, and training with Sharpness-Aware optimizers. In table 5 we verify the importance of these choices. We repeat the experiments done in section 5.1, ablating each of the three changes. We report results fine-tuning individual pretrained WRN34-10 models. We distill dataset with the same size as the original. As seen in table 5, all changes result in performance increases, with the largest being the use of the class rescaling technique, showing the importance of correctly tuning the noise and clipping. Using SAM optimizers affords the second greatest performance gain. As Sharpness-Aware optimizers have been shown to be robust to data noise (Foret et al., 2021), it makes sense that these optimizers will improve performance when training datasets created using noisy statistics.

Interestingly each of these changes affect different parts of the distillation to validation pipeline. The use of SiLU over ReLU activations is a design choice made during the pre-training stage of the model, the class rescaling technique only affects the actual distillation algorithm, and the optimizer choice at validation only affects the very last step of the pipeline. This highlights the complexity of private training pipelines, and suggests many different areas to improve distillation-based privacy techniques.

Table 6: SPS+ performance on CIFAR-10 classification at varying $D_C, D_G$ values at $\epsilon = 8$. * indicates the default value.

| $[D_C, D_G]$ | [256,256] | [384,384] | **[512,512]**$^*$ | [640,640] |
|---|---|---|---|---|
| WRN34-10 | $96.0 \pm 0.3$ | $96.3 \pm 0.2$ | $96.6 \pm 0.1$ | $96.4 \pm 0.2$ |
| WRN34-10 (Ensemble) | 96.7 | 96.9 | 97.2 | 97.2 |

Table 7: SPS+ performance on CIFAR-10 classification at varying $L_C$ values at $\epsilon = 8$. * indicates the default value used in the paper. * indicates the default value

| $L_C$ | [18] | [17,18] | **[16,17,18]**$^*$ | [15,16,17,18] | [14,15,16,17,18] |
|---|---|---|---|---|---|
| WRN34-10 | $96.0 \pm 0.1$ | $96.3 \pm 0.2$ | $96.6 \pm 0.1$ | $96.5 \pm 0.2$ | $96.6 \pm 0.1$ |
| WRN34-10 (Ensemble) | 96.8 | 97.1 | 97.2 | 97.2 | 97.2 |

### B.4 SPS VS SPS+ ABLATIONS

In this section we show the effect of sequentially adding grouped-pseudoclasses (GPC) and multi-stage clipping (MC) to SPS to get SPS+ in table 8. We see that GPC offers a very large benefit, particularly for many-class tasks, and MC gives a greater benefit for fewer class tasks.

### B.5 EFFECT OF $D_C, D_G$ AND $L_C$

Hyperparameter tuning in differentially private learning usually requires training many models on the same sensitive data, and each run consumes part of the total privacy budget (Papernot & Steinke, 2022). Therefore, a training algorithm which is sensitive to hyperparameters or requires extensive tuning, is riskier from a privacy standpoint. Here we study the effect of varying the $[D_C, D_G]$ parameteers and the $L_C$ parameters on CIFAR-10 training at $\epsilon = 8$.

table 6 and table 7 show that SPS+ remains fairly robust to these hyperparameters. Due to the fixed privacy budget, the noise added to each parameter scales inversely with the parameter count: for larger $L_C$ or $D_C/D_G$ values, the noise added is larger, compensating for receiving more statistics about the dataset. Consequently, SPS' performance is generally stable with respect to these hyper-parameters.

### B.6 CENTRALIZED GENERATION FEDERATED LEARNING WITH SPS

section 5.5 focuses on federated SPS where each client independently generated distilled datasets, and send them to a central server for aggregation. However, when client datasets are very small or highly non-IID, locally trained generators may produce lower-quality synthetic data. SPS naturally supports a variant that addresses this by performing **centralized generation** while still keeping only privatized statistics on the server.

Concretely, instead of generating synthetic data on each client, client $i$ computes its privatized summary statistics $\tilde{v}^{(i)}$ (as in our standard SPS pipeline) and sends $\tilde{v}^{(i)}$ to the server. The server then aggregates these summaries, for example via a dataset-size–weighted average: $\tilde{v}^{\text{combined}} = \sum \alpha_i \tilde{v}^{(i)}$, where $\alpha_i \propto |\mathcal{X}_T^i|$. Because each of $\tilde{v}^{(i)}$ are privatized, this still satisfies the DP requirements of each client.

The server can then generate a single synthetic dataset $\mathcal{X}_S$ from $\tilde{v}^{\text{combined}}$ and train the downstream model on $\mathcal{X}_S$ This centralized-generation variant allows the generator to effectively see a DP-sanitized approximation of the union of all clients' data, which directly mitigates issues arising from very small or highly imbalanced individual clients.

In the current paper we focus on the **decentralized generation** variant described in the previous comment, where each client generates its own privatized synthetic set and these are combined at the server. A detailed empirical comparison between decentralized-generation and centralized-generation SPS in extreme non-IID regimes is an interesting direction that we view as future work.

Table 8: Effect of removing multistage clipping and grouped pseudo-classes from SPS+, evaluated on WRN 28-10s.

| Method | CIFAR-10 | | | | CIFAR-100 | | | |
|---|---|---|---|---|---|---|---|---|
| | $\epsilon = 1$ | $\epsilon = 2$ | $\epsilon = 4$ | $\epsilon = 8$ | $\epsilon = 1$ | $\epsilon = 2$ | $\epsilon = 4$ | $\epsilon = 8$ |
| SPS | $93.2 \pm 0.2$ | $94.6 \pm 0.2$ | $95.0 \pm 0.2$ | $95.9 \pm 0.1$ | $48.9 \pm 1.1$ | $54.0 \pm 1.0$ | $66.3 \pm 0.4$ | $70.7 \pm 0.4$ |
| SPS+ (✗MC) | $94.4 \pm 0.3$ | $95.4 \pm 0.1$ | $95.7 \pm 0.0$ | $96.0 \pm 0.1$ | $70.1 \pm 0.6$ | $72.4 \pm 0.4$ | $73.7 \pm 0.4$ | $75.3 \pm 0.2$ |
| SPS+ | $\mathbf{95.1 \pm 0.3}$ | $\mathbf{95.9 \pm 0.1}$ | $\mathbf{96.3 \pm 0.1}$ | $\mathbf{96.3 \pm 0.2}$ | $\mathbf{71.0 \pm 0.3}$ | $\mathbf{74.3 \pm 0.3}$ | $\mathbf{76.2 \pm 0.3}$ | $\mathbf{77.5 \pm 0.1}$ |

# C  BACKGROUND ON APPROXIMATE DP, RENYI DIFFERENTIAL PRIVACY AND PROOF OF THEOREM 4.1

A more classic and also commonly-used DP definition is approximate DP Dwork et al. (2006) whose security parameters $(\epsilon, \delta)$ are formally defined as follows.

**Definition C.1** (Approximate Differential Privacy Dwork et al. (2006)). *Given a universe $\mathcal{X}$, we say that two datasets $X, X' \subseteq \mathcal{X}$ are adjacent, denoted as $X \sim X'$, if $X = X' \cup \{x\}$ or $X' = X \cup \{x\}$ for some additional datapoint $x \in \mathcal{X}$. A randomized algorithm $\mathcal{M}$ is said to be $(\epsilon, \delta)$-differentially private (DP) if for any pair of adjacent datasets $X, X'$ and any event set $O$ in the output domain of $\mathcal{M}$, it holds that*

$$\mathbb{P}(\mathcal{M}(X) \in O) \leq e^{\epsilon} \cdot \mathbb{P}(\mathcal{M}(X') \in O) + \delta.$$

Interpreted through binary hypothesis testing—where an adversary attempts to distinguish between two adjacent datasets, $X$ and $X'$—small values of $\epsilon$ and $\delta$ imply that either the Type-I or Type-II error must be large Kairouz et al. (2015). Another fundamental concept in differential privacy is *composition*. Additionally, the conversion from RDP to $(\epsilon, \delta)$-DP is characterized in the following lemma.

**Lemma C.2** (Advanced Composition via RDP and Conversion Mironov (2017b)). *For any $\alpha > 1$ and $\epsilon > 0$, the class of $(\alpha, \epsilon(\alpha))$-RDP mechanisms satisfies $(\tilde{\epsilon}, \tilde{\delta})$-differential privacy under $T$-fold adaptive composition for any $\tilde{\epsilon}$ and $\tilde{\delta}$ such that*

$$\tilde{\epsilon} = T\epsilon(\alpha) - \log(\tilde{\delta})/(\alpha - 1).$$

Also, Gaussian noise, as a popular randomization, is known to produce required RDP guarantees once sensitivity is known:

**Lemma C.3** (Gaussian Mechanism RDP Mironov (2017b)). *Suppose a processing function $\mathcal{F}$ whose sensitivity is bounded by $|v|$ in $L_2$ norm, i.e., $\|\mathcal{F}(X) - \mathcal{F}(X')\| \leq |v|$ for arbitrary two adjacent datasets $X \sim X'$, then for a Gaussian noise sampled from $\mathcal{N}(0, \sigma^2 \cdot \boldsymbol{I})$, the noisy mechanism $\mathcal{M}(\cdot) = \mathcal{F}(\cdot) + e$, i.e., the output $\mathcal{F}$ independently perturbed by $e$, satisfies $(\alpha, \alpha/2 \cdot |v|^2/\sigma^2)$-RDP.*

## C.1  PROOF OF SPS PRIVACY

We restate and provide proof of theorem 4.1 here.

**Theorem 4.1** (Privacy of SPS). *The release of $\tilde{v}$ in eq. (4) for M models satisfies $(\alpha, \epsilon)$-RDP, where $\epsilon = \frac{M\alpha}{2b_0^2}$ for $\alpha > 1$. Proof. See section C.1* $\square$

*Proof.* From lemma C.3, by clipping, the sensitivity of the aggregation is bounded by $|v|_{\max}$ in $L_2$ norm, and thus each release of $\tilde{v}$ for each model is $(\alpha, \frac{\alpha}{2b_0^2})$-RDP. Using lemma 2.2, we have that the total composed mechanism is $(\alpha, \frac{M\alpha}{2b_0^2})$-RDP. $\square$

# D  IMPLEMENTATION DETAILS

## D.1  PRETRAINED MODELS

We use pretrained models of architectures WRN22-8, WRN28-10, WRN34-10 trained on $32 \times 32$ downsampled ImageNet. For the model used using synthesis, we using SiLU activations, and for validation we use ReLU activations. We train for 90 epochs with SGD optimizer and a Cosine learning rate schedule, and a linear warmup of 5 epochs. These models achieve $63\%$ on ImageNet $32 \times 32$.

Table 9: Hyperparameters set constant for all CIFAR-10 and CIFAR-100 Experiments. We use $L_C = [15, 16, 17, 18, 19]$ for CIFAR-100 $\epsilon = 8$, SPS+ as the only exception. SPS+ uses a different clipping factor for each stage, reducing the clipping radius each time.

| Hyperparameter | CIFAR-10 (SPS) | CIFAR-100 (SPS) | CIFAR-10 (SPS+) | CIFAR-100 (SPS+) |
|---|---|---|---|---|
| $K_{\text{clip}}$ | $\frac{1}{8}$ | $\frac{1}{16}$ | [16,16,32,32,32] | [16,16,32,32,32] |
| $\gamma_{\text{eig}}$ | 2.0 | 2.0 | 2.0 | 2.0 |
| $L_C$ | [17,18,19] | [17,18,19] | [17,18,19] | [17,18,19] |
| $\lambda_C$ | 30 | 30 | 30 | 30 |

Table 10: $[D_G, D_C]$ for different CIFAR-10 and CIFAR-100 experiments

| Privacy Budget | CIFAR-10 (SPS) | CIFAR-100 (SPS) | CIFAR-10 (SPS+) | CIFAR-100 (SPS+) | CIFAR-10 Federated (SPS+) | CIFAR-100 Continual (SPS+) |
|---|---|---|---|---|---|---|
| $\epsilon = 1$ | [192,192] | [128,32] | [192,192] | [96,96] | [128,32] | [128,32] |
| $\epsilon = 2$ | [256,256] | [128,32] | [256,256] | [128,128] | [160,48] | [160,48] |
| $\epsilon = 4$ | [512,512] | [256,48] | [384,384] | [192,192] | [192,64] | [192,64] |
| $\epsilon = 8$ | [512,512] | [512,64] | [512,512] | [256,256] | [256,128] | [256,128] |

## D.2 SYNTHESIS HYPERPARAMETERS

table 9 Shows the hyperparameters which were set at constant values for all CIFAR-10/CIFAR-100 experiments. We note that $[17, 18, 19]$ represents the last three layers of the WRN22-8 model used during distillation, which contains 19 BatchNorm layers.

Table 10 shows the choices of $D_G$ and $D_C$ in different experiments. These parameters were not tuned and chosen rather arbitrarily. It is very likely better performance can be gained by picking better ones. In general for larger privacy budgets, small $D_G$ and $D_C$ should be used. When there are more classes, the ratio of $D_G$ to $D_C$ should be larger.

For the sake of transparency, we also show the values of $b_0$ use for each privacy budget and model count used during distillation in table 11. These values were calculated using RDP accountant for a Gaussian Mechanism with no subsampling, using dp_accounting library at https://github.com/google/differential-privacy. This implements a numerical procedure which is detailed in proposition 12 in Canonne et al. (2020).

When synthesizing images, optimize them for 1000 iterations of Adam optimizer with initial learning rate of 0.25, $\beta_1 = 0.5, \beta_2 = 0.9$, and a batch size of 200. For CIFAR-100 this is 2 images per class per batch and for CIFAR-10 it is 20. For data augmentation we use the same linear data augmentation curriculum presented in Yin & Shen (2023), with a random horizontal flip and random resized crop with minimum size 0.3. Images are initialized as random noise mean 0 and standard deviation 1. We additionally apply a small penalty for pixel values which are outside of the range of the permissible RGB levels. Additionally, we adopt of the EMA statistic accumulation method proposed in Loo et al. (2024) optimize EMA values for $\mu_l^{G/c}$.

### D.2.1 SPS+ HYPERPARAMETERS

For SPS+, we use a learning rate of 0.1 for stages after the first one, and optimize for only 500 iterations, initialize with the previous stage's images. We use $K = 40, 200$ for CIFAR-10 and CIFAR-100, respectively, with $N_{c/p} = 2, 10$. This leads to 10000 images per pseudoclass for CIFAR-10, and 5000 per pseudoclass for CIFAR-100. For class-incremental CIFAR-100, we use $K = 40, N_{c/p} = 2$. $K_{\text{clip}}$ is varied at each stage, going from $K_{\text{clip}}^1 = 16$ to $K_{\text{clip}}^5 = 32$, gradually decreasing the clipping ratio, as per table 9.

### D.3 VALIDATION

When validating on pre-trained WRN models, we use GSAM optimizer based on Adam with learning rate $1e - 5$ and batch size of 256. For the GSAM optimizer, we use $\alpha = 0.04$, and use a proportional $\rho$ schedular with max values 0.05 and min value 0.02. We use the non-adaptive variant of GSAM. Addtionally, we freeze batchnorm layers. We train for epochs shown in table 12

Table 11: $b_0$ noise cofficient used for different privacy budgets and model counts. Values are calculated using the `dp_accounting` library

| Privacy Budget | $M = 1$ | $M = 2$ | $M = 3$ | $M = 4$ | $M = 5$ |
|---|---|---|---|---|---|
| $\epsilon = 1$ | 4.045 | 5.720 | 7.006 | 8.090 | 9.045 |
| $\epsilon = 2$ | 2.149 | 3.039 | 3.722 | 4.298 | 4.805 |
| $\epsilon = 4$ | 1.157 | 1.637 | 2.004 | 2.315 | 2.588 |
| $\epsilon = 8$ | 0.637 | 0.901 | 1.104 | 1.275 | 1.425 |

Table 12: Epochs trained during validation

| Dataset Size | 1000 | 2500 | 5000 | 10000 | 15000 | 20000 | 25000 | 40000 | 50000 |
|---|---|---|---|---|---|---|---|---|---|
| Epochs Trained | 100 | 60 | 45 | 30 | 24 | 20 | 18 | 12 | 10 |

### D.4 HARDWARE AND RUNTIME

All experiments were run on NVIDIA H100s with 80GB of VRAM. Generating a batch of 200 images takes between 2-5 minutes for 1000 optimization steps, depending on the configuration of $D_G$ and $D_S$ used. This means that generating a dataset of size 50000 takes between 8-21 hours on a single NVIDIA H100. The synthesization step can be done in parallel on multiple GPUS. SPS+ takes $1 + \frac{M-1}{2}$ times as long as SPS, as each subsequent stage is optimized for half as many optimization steps.

## E BASELINES FOR FEDERATED LEARNING

In section 5.5, we compared SPS to two data-base federated learning baselines: FedDM (Xiong et al., 2022), and FedLAP-DP (Wang et al., 2024). In this section we give a brief description of each and detail how we implemented as a baseline. We did hyperparameter tuning for both methods.

**FedDM** (Xiong et al., 2022) is a federated version of distribution matching (DM), a common Dataset Distillation algorithm. It works by matching the mean embedding of features at each federated learning checkpoint. This method is **not differentially private**, and thus is an unfair comparison to ours. Typically, FedDM initializes each set of synthetic images with real images, and modifies them during the optimization procedure, then sends the modified images. In order to make the comparison to a private approach more fair, **we changed the initialization from real images to random noise**, as an private algorithm cannot use real images as the initialization. Like our method, we use the same WRN34-10.

**FedLAP-DP** (Wang et al., 2024) works by essentially matching gradients in the local area around the federated learning checkpoint. These gradients are privatized using the standard DP-SGD clipping/noising procedure. This means that the privacy cost composes over each synthetic image update step. We note that the original code repository incorrectly computes the privacy budget only within each communication round, and does not compose over multiple. We have corrected this in our reported number.

We note that our method also cannot be directly compared to (Xiong et al., 2022) and (Wang et al., 2024), as theirs uses multiple rounds of communication between the client and servers, while ours just uses one at the start. Furthermore, our method also does not use knowledge of the original checkpoint (for example we distill using a WRN22-8, and evaluate on a WRN34-10).

## F COMPARISON TO EXISTING DIFFERENTIALLY PRIVATE IMAGE GENERATION METHODS

In this section we compare to compare to more DP-generation or DP-SGD based works on CIFAR-10 classification. table 13 shows the results. Unless otherwise stated, the public dataset is ImageNet1k, and $\delta = 1e - 5$. We note that generally speaking, gradient-based methods which leverage

DP-SGD perform better than generation based methods, with the exception of SPS. Additionally, methods which try to directly optimize in pixel space, which tend to be motivated from the dataset distillation method, with the exception of SPS (Vinaroz & Park, 2024; Chen et al., 2022), perform very poorly compared to methods which leverage diffusion models.

We note that we were unable to compare to DP-MERF (Harder et al., 2020), because their method only was validated on the much simpler MNIST/FashionMNIST datasets, and achieves $< 90\%$ on both datasets, making the comparison unnecessary. Despite this, both methods are similar in the sense that their aim to match distribution wide statistics, however ours uses a more refined choice of statistics to match.

We note that DP-KIP, Vinaroz & Park (2024) typically evaluates their methods using Kernel Ridge Regression (KRR), rather than directly training a model on the generated images using SGD. When reproducing their results, we also training on their distilled datasets using SGD, which results in 10% random guessing performance. We believe that this is because DP-KIP tends to produces images that look like random noise (see figure 2 in (Vinaroz & Park, 2024)), which do not generalize well. This random noise look is partially due to the limited number of update steps which can be used on the generated images, due to the growing privacy cost due to composition. In contrast, our method's privacy cost does not grow with the number of optimization steps of the images, allowing us to use thousands to steps per image to produce better looking images as shown in fig. 4.

While SPS excels in terms of accuracy, SPS is not competitive with methods that use pre-trained generators (DP-Diffusion (Dockhorn et al., 2022; Ghalebikesabi et al., 2023) and Private Evolution(Lin et al., 2024)) in terms of FID score. Compared to methods which do not use pretrained generators (Private Set Generation, DP-KIP), SPS achieves much better FID scores. This discussion will make its way into the paper.

Table 13: Accuracy of various DP classification methods on CIFAR-10 in terms of downstream accuracy and FID (for generative methods).

| Method Type | Method | Accuracy (%) | FID |
|---|---|---|---|
| Gradient-based | DP-RandP ((Tang et al., 2023)) ($\epsilon = 8$, no public data) | 85.26 | – |
| | Mehta et al. (2022) ($\epsilon = 8$) | 91.3 | – |
| | Bu et al. (2022) ($\epsilon = 8$) | 96.5 | – |
| | De et al. (2022) ($\epsilon = 8$) | 96.6 | – |
| Generative | DP-Diffusion (Ghalebikesabi et al., 2023) ($\epsilon = 10$) | 88.0 | 9.8 |
| | Chen et al. (2022) ($\epsilon = 10$) | 42.6 | 444.6 (reproduced) |
| | Private-Evolution (Ghalebikesabi et al., 2023) ($\epsilon = 10$) | 89.13 | <7.9 |
| | DP-KIP (Vinaroz & Park, 2024) ($\epsilon = 10$, KRR, no public data) | 58.7 | Not reported |
| | DP-KIP (Vinaroz & Park, 2024) ($\epsilon = 8$, KRR, no public data) (reproduced) | 53.9 | 423.5 (reproduced) |
| | DP-KIP (Vinaroz & Park, 2024) ($\epsilon = 8$, SGD) (reproduced) | 10.0 | 423.5 |
| | **Ours, SPS** ($\epsilon = 8$) | 96.1 | 25.0 |
| | **Ours, SPS, Ensemble** ($\epsilon = 8$) | 96.8 | 25.0 |
| | **Ours, SPS+** ($\epsilon = 8$) | 96.6 | 22.5 |
| | **Ours, SPS+, Ensemble** ($\epsilon = 8$) | **97.2** | 22.5 |

## F.1 RUNTIME COMPARISON WITH DP-SGD

In this section we detail the compute requirements compared to DP-SGD. While we cannot directly reproduce the experiments in (De et al., 2022), due to the large resource requirements, we are able to estimate the compute costs. From figure 7 (page 16), a batch size of 256 requires approximately 32k training iterations. On an H100 training on a WRN 34-10 (to be comparable with our paper), a single privatized gradient step with batch size 256 takes 0.32s. This gives 2.8 hours on an H100. However, this is the results with augmentation multiplicity of 1. The experiments in (De et al., 2022) report using an augmentation multiplicity $> 16$, which directly scales the batch size (and time taken is linear with batch size). This means to get comparable results, we would need approximately 45 H100 GPU hours. This is comparable to the experiments in this work which take around 8-20 (see section D.4). We note that improving the efficiency of DP-generation methods is a promising direction for future work.

Table 14: Oversized synthesis performance of SPS+ on CIFAR-100 for WRN34-10 ensembles. Oversized synthetic datasets can further improve performance

| | Distilled Dataset size | | | | |
|---|---|---|---|---|---|
| $\epsilon$ | 1× | 2× | 3× | 4× | DP-SGD |
| 1 | **76.6** | 76.4 | 76.0 | 75.9 | 70.3 |
| 2 | 79.2 | **79.4** | **79.4** | 79.3 | 74.7 |
| 4 | 80.7 | 81.2 | **81.3** | 81.1 | 79.2 |
| 8 | 81.6 | 81.8 | **82.1** | 81.9 | 81.8 |

Table 15: Oversized synthesis performance of SPS+ on CIFAR-10 for WRN34-10 ensembles.

| | Distilled Dataset size | | | | |
|---|---|---|---|---|---|
| $\epsilon$ | 1× | 2× | 3× | 4× | DP-SGD |
| 1 | 96.2 | 96.0 | 96.0 | 95.7 | 94.8 |
| 2 | 96.8 | 96.5 | 96.5 | 96.5 | 95.4 |
| 4 | 97.1 | 97.0 | 96.9 | 96.9 | 96.1 |
| 8 | 97.2 | 97.2 | 97.1 | 97.2 | 96.6 |

# G ADDITIONAL RESULTS

## G.1 TINY-IMAGENET

In this work, we focused on lower resolution data such as CIFAR-10, and CIFAR-100. This was mainly due to the constraint of requiring a larger public dataset for pretraining that is disjoint from the private dataset. De et al. (2022) were able to leverage proprietary datasets such as JFT-300M/JFT-4B for their pretraining datasets, which we do not have access to. To validate SPS+ at higher resolution, more complex datasets, we ran additional experiments on Tiny-ImageNet (64x64, 100k samples, 200 classes) at $\epsilon = 8$ with SPS+. For the pretraining dataset we used a 64x64 variant of ImageNet-1K, with overlapping classes with Tiny-ImageNet removed, leading to a 818-class pretraining dataset. At $\epsilon = 8, \delta = 4e-6$, we achieve $49.5\%$ accuracy with an ensemble of WRN28-10 models, using a WRN22-8 model for distillation.

## G.2 OTHER EXPERIMENTS

table 15 and table 14 show the performance of distilling synthetic datasets larger than the original dataset. For CIFAR-100, oversized distillation can improve performance, but there is no major effect for CIFAR-10.

Fig. 6 shows additional results to supplement fig. 2, including the WRN22-8 results. Fig. 7 has more results to supplement fig. 5. Fig. 8 shows the similar continual learning results as fig. 5, but with more trained models, and fig. 9 shows more models trained on the same federated version of CIFAR-10 as fig. 5.

# H VISUALIZING DISTILLED IMAGES

Distilled images for the CAMELYON-17 dataset are in fig. 10 .We show more distilled images in fig. 11 and fig. 12 for CIFAR-10 and CIFAR-100, respectively

# I LLM USAGE

LLMs were used for editing the final draft of the paper.

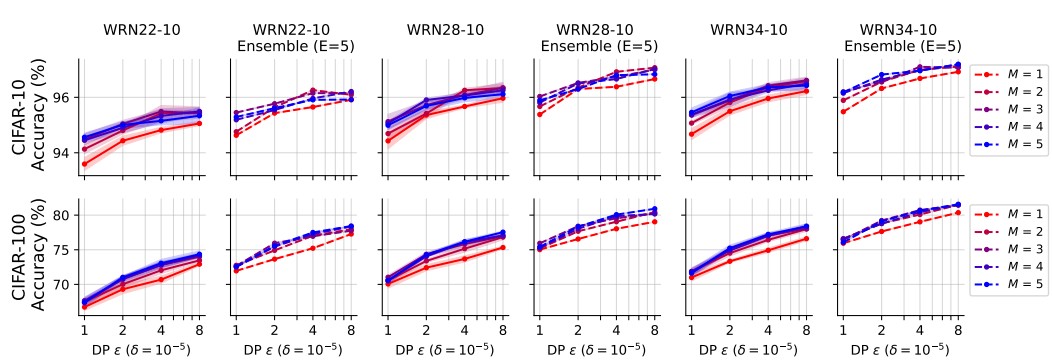

Figure 6: Accuracy of pretrained Wide ResNet models fine tuned on SPS synthesized images on CIFAR-10 and CIFAR-100.

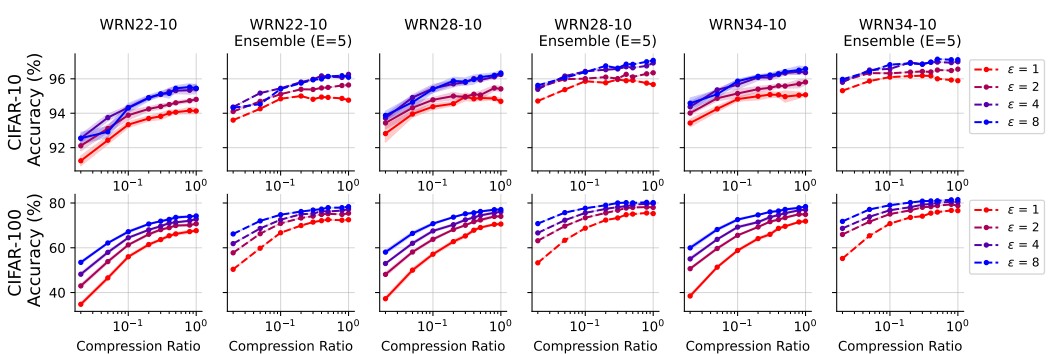

Figure 7: Simulataneous privatization and distillation for various fine-tuning models and privacy budgets

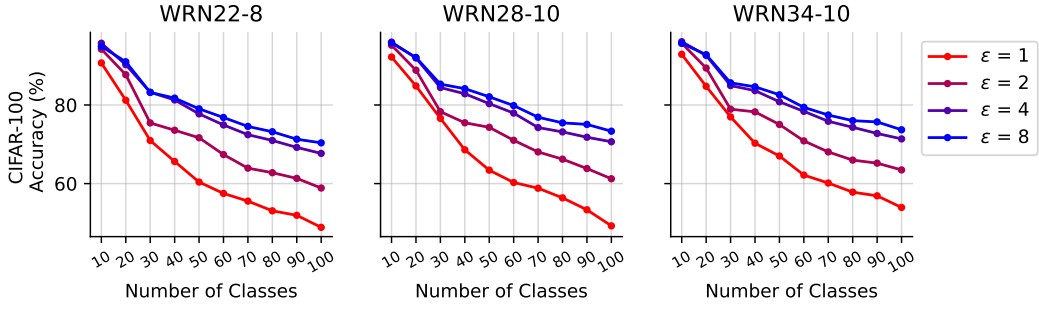

Figure 8: Class incremental CIFAR-100 trained on more models

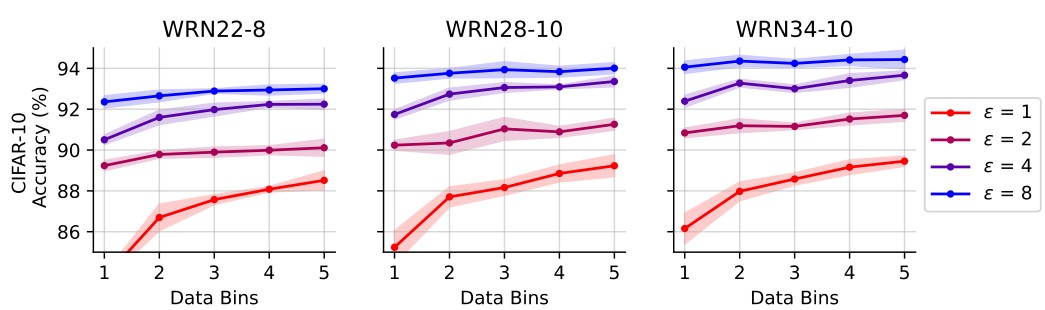

Figure 9: Federated CIFAR-10 trained on more models

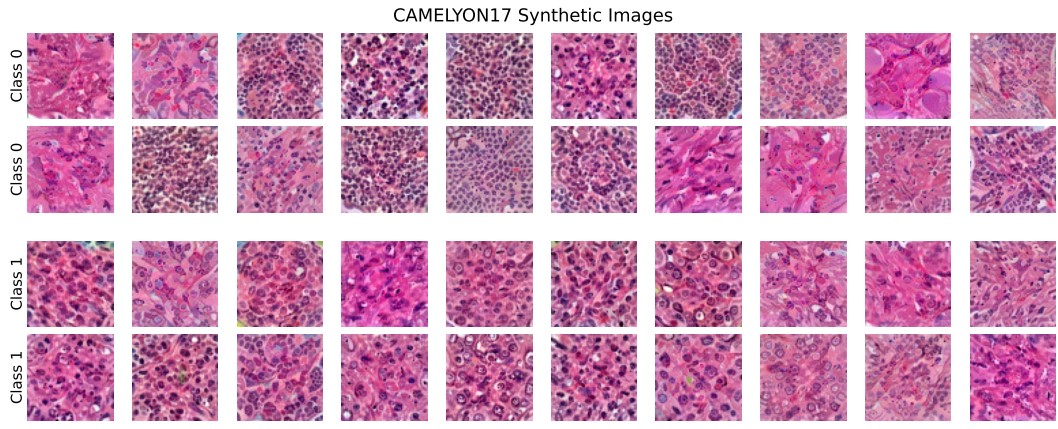

Figure 10: Distilled Images on CAMELYON17 ($\epsilon = 8$)

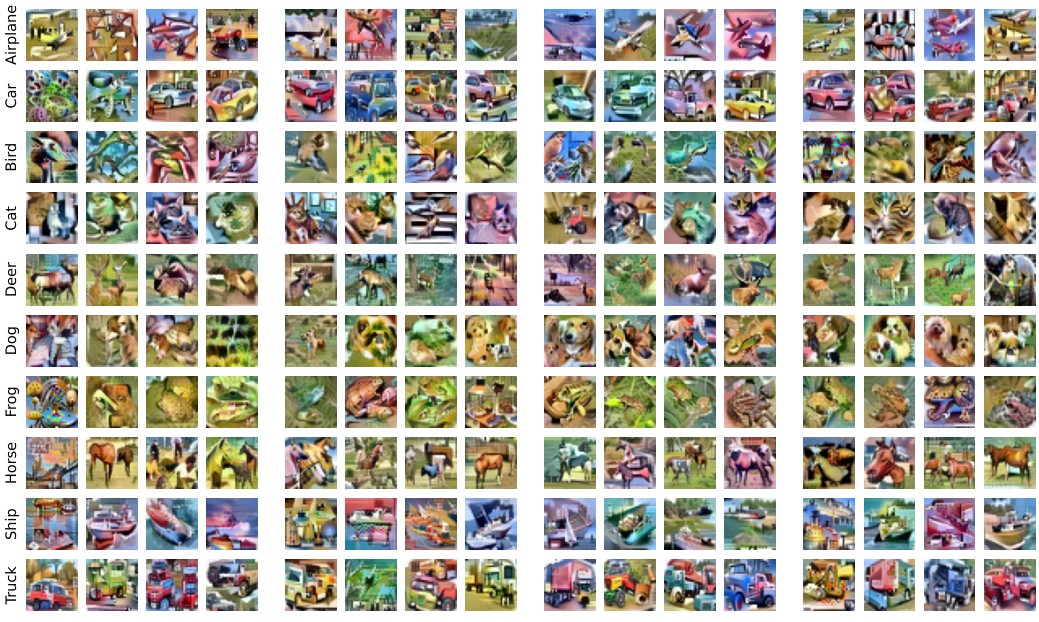

Figure 11: Distilled Images on CIFAR-10

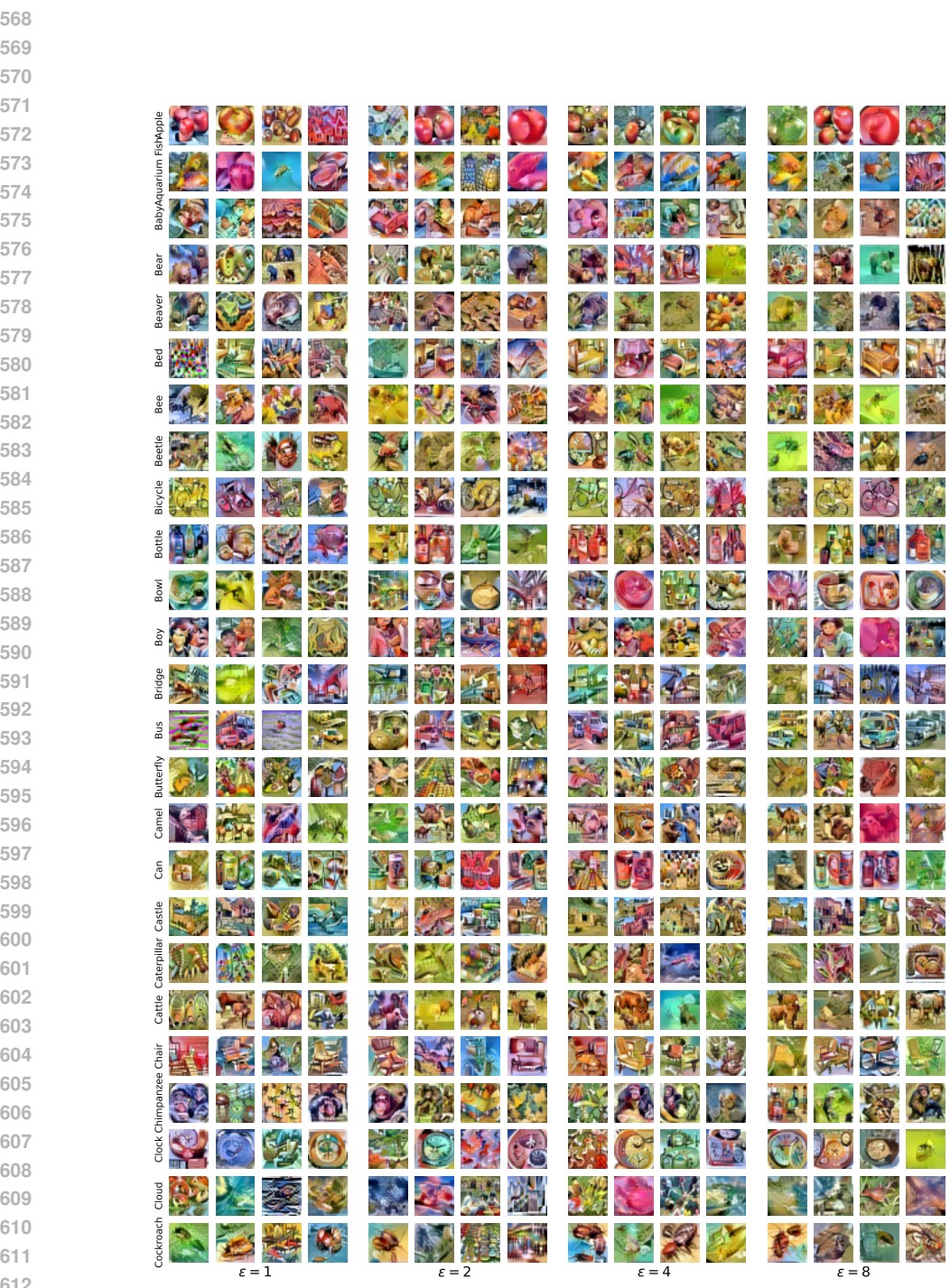

Figure 12: Distilled Images on CIFAR-100

