# OpenReview forum: "High Performance Differentially Private Fine-Tuning using Dataset Distillation"
_ICLR.cc/2026/Conference — Submitted to ICLR 2026_

### Official Review · Reviewer_UvM8 · 2025-10-16

**Soundness:** 3
**Presentation:** 3
**Contribution:** 2
**Rating:** 4
**Confidence:** 3

**Summary:**

This paper integrates differential privacy into the D3S data synthesis framework and evaluates the utility of the resulting private synthetic data on downstream training tasks.

**Strengths:**

The paper effectively demonstrates that models trained on the private synthetic data can achieve better accuracy compared to those trained with DP-SGD on the real data.

**Weaknesses:**

The comparison is primarily against DP-SGD. To properly situate the contribution, it is crucial to also benchmark against other state-of-the-art private data synthesis methods on both (1) statistical fidelity of the data and (2) utility on downstream tasks.

**Questions:**

1. Figure 3 are difficult to interpret. A quantitative evaluation would provide a more clear measure of image quality across different privacy budgets.

2. In Figure 4, the utility increase as the compression ratio increases. What happens when the ratio is >1? Does performance also increase, or does it simply plateau? This is important for understanding the method's limits.

---

> ### Author Response · Authors · 2025-11-18
>
> We appreciate the reviewers suggestions on how to improve the paper and aim to address their concerns here.
>
> **Comparison to other data synthesis methods.** As mentioned in lines 393-394, in Table 1, we only report the best performing DP-synthesis method, Private Evolution due to space constraints. In Table 10 in Appendix F we compare to other synthesis based approaches. We summarize table F below for the reviewer’s convenience.
>
> | Method                                                                                         | Accuracy  |
> |------------------------------------------------------------------------------------------------|-------:|
> | DP-Diffusion (Ghalebikesabi et al., 2023) (ε = 10)                                             | 88.0   |
> | Chen et al. (2022) (ε = 10)                                                                    | 42.6   |
> | Private-Evolution (Ghalebikesabi et al., 2023) (ε = 10)                                        | 89.13  |
> | DP-KIP (Vinaroz & Park, 2024) (ε = 10, KRR, no public data)                                    | 58.7   |
> | DP-KIP (Vinaroz & Park, 2024) (ε = 8, KRR, no public data) (reproduced)                        | 53.9   |
> | DP-KIP (Vinaroz & Park, 2024) (ε = 8, SGD) (reproduced)                                        | 10.0   |
> | **Ours, SPS (ε = 8)**                                                                          | 96.1   |
> | **Ours, SPS, Ensemble (ε = 8)**                                                                | 96.8   |
> | **Ours, SPS+ (ε = 8)**                                                                         | 96.6   |
> | **Ours, SPS+, Ensemble (ε = 8)**                                                               | **97.2** |
>
>
>
> We see very clearly that SPS significantly outperforms other methods, even when competing methods have higher privacy budgets. Notably, SPS represents **the first instance where generation-based methods compete with/outperform gradient-based methods in private image classification**
>
> **Figure 3 interpretation.** To better quantify the increase in image quality, we will modify the image to include the FID scores of the generated images in figure 3. We summarize these numbers below:
>
> | Privacy Budget | $\epsilon = 1$ | $\epsilon = 2$ | $\epsilon = 4$ | $\epsilon = 8$ |
> |----------------|----------------|----------------|----------------|----------------|
> | FID Score      | 42.00          | 36.71          | 32.13          | \textbf{24.36}          |
>
> The main takeaway is that higher privacy budgets also result in visually identifiable images.
>
> We will additionally include a figure showing the FID score as function of privacy budget $\epsilon$ and stage count $M$ in the rebuttal revision, and will update the reviewer once this revision is ready. Accuracy of these images at different privacy budgets is present in table 1 and figure 2.
>
> **SPS beyond compression ratio 1.** This is a very interesting point raised by the reviewer. We ran additional experiments of SPS+, creating datasets up to 4x larger than the original. We show their performance below:
>
> We see that particularly for CIFAR-100, we can achieve even better performance with oversized distilled datasets.
>
> Table 1: CIFAR-100 SPS+ performance at oversized distilled dataset sizes on WRN34-10 ensembles
> | Distilled Dataset size | 1x       | 2x       | 3x       | 4x   | DP-SGD |
> |------------------------|----------|----------|----------|------|--------|
> | $\epsilon = 1$         | **76.6** | 76.4     | 76.0     | 75.9 | 70.3   |
> | $\epsilon = 2$         | 79.2     | **79.4** | **79.4** | 79.3 | 74.7   |
> | $\epsilon = 4$         | 80.7     | 81.2     | **81.3** | 81.1 | 79.2   |
> | $\epsilon = 8$         | 81.6     | 81.8     | **82.1** | 81.9 | 81.8   |
>
> Table 1: CIFAR-10 SPS+ performance at oversized distilled dataset sizes on WRN34-10 ensembles
>
> | Distilled Dataset size | 1x   | 2x   | 3x   | 4x   | DP-SGD |
> |------------------------|------|------|------|------|--------|
> | $\epsilon = 1$         | 96.2 | 96.0 | 96.0 | 95.7 | 94.8   |
> | $\epsilon = 2$         | 96.8 | 96.5 | 96.5 | 96.5 | 95.4   |
> | $\epsilon = 4$         | 97.1 | 97.0 | 96.9 | 96.9 | 96.1   |
> | $\epsilon = 8$         | 97.2 | 97.2 | 97.1 | 97.2 | 96.6   |
>
>
>
> **Other contributions.** We would also like to point out other contributions of this work which were not addressed by the reviewer, namely SPS’s application to federated learning and continual learning. SPS adapts without modification to both settings, whereas DP-SGD, for example, fails in continual learning due to the catastrophic forgetting phenomenon.
>
> These new results will make their way into the main text and we will notify the reviewer once these modifications are complete.

---

> > ### Comment · Reviewer_UvM8 · 2025-11-19
> > **Comment**
> >
> > Thanks for your response, it addresses part of my concern.
> >
> > 1. Utility and fidelity: As noted earlier, accuracy is ultimately a measure of downstream utility. I  believe it would be beneficial to include an evaluation of the fidelity of the synthetic data itself. Since the experiments involve images, the FID score seems like an appropriate choice.
> > 2. Federated and Continual Learning: You mention advantages in federated learning and continual learning that DP-SGD does not provide. However, it seems to me that *any* private data synthesis method would share these advantages, not just the proposed approach. Do you agree?

---

> ### Author Response · Authors · 2025-11-21
>
> Thank you for the response. The reviewer raises some interesting points we'd like to discuss here.
>
> **Utility and Fidelity. ** As requested by the reviewer, we report the FID scores of these methods below. We note that some methods do not directly state the FID score at comparable $\epsilon$ values, but we report the closest one where available.
>
> Table 1: Downstream accuracy and FID of private generative methods
>
> | Method                                                                                         | Accuracy | FID |
> |------------------------------------------------------------------------------------------------|---------:|----:|
> | DP-Diffusion (Ghalebikesabi et al., 2023) (ε = 10)                                             |    88.0  |  9.8 |
> | Private Set Generation (Chen et al. 2022) (ε = 10)                                                                    |    42.6  |  444.6 (reproduced) |
> | Private-Evolution (Ghalebikesabi et al., 2023) (ε = 10)                                        |   89.13  |  **<7.9** (score for ε = 0.67 reported) |
> | DP-KIP (Vinaroz & Park, 2024) (ε = 10, KRR, no public data)                                    |    58.7  | Not reported |
> | DP-KIP (Vinaroz & Park, 2024) (ε = 8, KRR, no public data) (reproduced)                        |    53.9  |  423.5 (reproduced) |
> | DP-KIP (Vinaroz & Park, 2024) (ε = 8, SGD) (reproduced)                                        |    10.0  |  423.5 |
> | **Ours, SPS (ε = 8)**                                                                          |   96.1   |  25.0 |
> | **Ours, SPS, Ensemble (ε = 8)**                                                                |   96.8   | 25.0 |
> | **Ours, SPS+ (ε = 8)**                                                                         |   96.6   |  22.5 |
> | **Ours, SPS+, Ensemble (ε = 8)**                                                               | **97.2** |  22.5 |
>
> SPS is not competitive with methods that use pre-trained generators (DP-Diffusion and Private Evolution) in terms of FID score. Compared to methods which do not use pretrained generators (Private Set Generation, DP-KIP), SPS achieves much better FID scores. This discussion will make its way into the paper.
>
> **This is indeed a current limitation of SPS and we thank the reviewer for pointing this out.** SPS is tailored towards downstream fine-tuning accuracy, and FID score is secondary. However, future work could look at achieving high performance in both fields. One possible extension of SPS is getting SPS to leverage pre-trained generators like DP-Diffusion and Private evolution to both improve image realism, as well as improve optimization time.
>
> **Federated and Continual Learning.** The reviewer is correct in stating all DP-generation methods can perform federated and continual learning. However, historically, these generation methods have had too low downstream accuracy to be viable for these use cases. Specifically, the tradeoff is that DP-SGD can achieve high accuracy in standard centralized training, but struggles when adapting to continual or federated learning. Data synthesis methods, on the other hand, can adapt, but are not viable because of their poor accuracy. SPS performs well in all these settings without modification.
>
> In figure 4e, for example, we show that SPS significantly outperforms FedLap-DP, a private federated learning algorithm based on data synthesis (>94% vs. <40%), and also FedDM, a non-private algorithm (<80%).

---

> > ### Comment · Reviewer_UvM8 · 2025-11-21
> > **Comment**
> >
> > Thanks for showing the results. Regarding the federated learning setting, the description is unclear to me.
> >
> > When you say “each party running SPS+ independently”, does this mean there is no aggregation during training? If so, how does the “distributed SPS+ effectively aggregate multiple data sources” as claimed?

---

> ### Author Response · Authors · 2025-11-21
>
> Let each of $N$ parties have a disjoint subset of the original dataset: ${\mathcal{X}_T^1, ... \mathcal{X}^N_T, }$. Each party run SPS+ locally and independently, generating synthetic privatized sets ${\mathcal{X}_S^1, ... \mathcal{X}^N_S}$. These are sent to a server, which combines the datasets into a single one: $\mathcal{X}_S^{combined} = \bigcup_i^N \mathcal{X}_S^i$, and trains on $\mathcal{X}_S^{combined}$, resulting in the accuracy shown in figure 4d/e.
>
> We say that distributed SPS+ effectively aggregates multiple data sources because we see that increasing the number of involved parties (when $|\mathcal{X}_T^i|$ is fixed, and N increases), improves performance, particularly for tight privacy budgets.
>
> We agree that the description in the main text is not particularly clear, as it was abridged in the main text due to the page limit. We will elaborate more in the rebuttal revision. We are open to suggestions from the reviewer on how we can make this section clearer.

---

> > ### Comment · Reviewer_UvM8 · 2025-11-24
> > **Comment**
> >
> > Non-aggregation federated methods face a challenge when client datasets are small or non-IID. This data heterogeneity can lead to poor quality in the generated data. Federated learning with aggregation could help resolve this problem.

---

> ### Author Response · Authors · 2025-11-25
>
> We thank the reviewer for raising this point. We agree that when client datasets are very small or highly non-IID, locally trained generators may produce lower-quality synthetic data. SPS naturally supports a variant that addresses this by performing **centralized generation** while still keeping only privatized statistics on the server.
>
> Concretely, instead of generating synthetic data on each client, client $i$ computes its privatized summary statistics $\tilde{v}^{(i)}$ (as in our standard SPS pipeline) and sends $\tilde{v}^{(i)}$ to the server. The server then aggregates these summaries, for example via a dataset-size–weighted average: $\tilde{v}^{combined} = \sum \alpha_i \tilde{v}^{(i)}$, where $\alpha_i \propto |\mathcal{X}^i_T|$. Because each of $\tilde{v}^{(i)}$ are privatized, this still satisfies the DP requirements of each client.
>
> The server then generates a single synthetic dataset $\mathcal{X}_S$ from $\tilde{v}^{combined}$ and trains the downstream model on $\mathcal{X}_S$ This centralized-generation variant allows the generator to effectively see a DP-sanitized approximation of the union of all clients’ data, which directly mitigates issues arising from very small or highly imbalanced individual clients.
>
> In the current paper we focus on the **decentralized-generation** variant described in the previous comment, where each client generates its own privatized synthetic set and these are combined at the server. Our goal in this section is primarily to demonstrate that SPS can be applied in federated settings without substantial modification. A detailed empirical comparison between decentralized-generation and centralized-generation SPS in extreme non-IID regimes is an interesting direction that we view as future work. If the reviewers find it helpful, we can include an additional experiment with centralized generation in the appendix to illustrate this variant.

---

> > ### Comment · Reviewer_UvM8 · 2025-11-25
> > **Comment**
> >
> > Thank you for the clarification. The method for aggregating the summary statistics now makes sense.

---

### Official Review · Reviewer_fbnc · 2025-10-30

**Soundness:** 3
**Presentation:** 2
**Contribution:** 3
**Rating:** 4
**Confidence:** 2

**Summary:**

This paper proposes a new DP synthetic data generation algorithm inspired from the data distillation literature.

Their method is named **SPS (Summarize-Privatize-Synthesize)**. It leverages a public, pre-trained model to:
1.  **Summarize** the private dataset by extracting intermediate activation statistics (class-conditional and global means and covariances).
2.  **Privatize** these statistics in a single step using the Gaussian Mechanism (by clipping and noising the sum of per-sample statistics).
3.  **Synthesize** a new dataset by optimizing it from random noise to match these privatized statistics.

The authors also introduce **SPS+**, an enhanced version that incorporates **Multistage Clipping (MC)** to iteratively refine the statistics and **Grouped Pseudo-classes (GPC)** to mitigate the high noise in many-class settings (like CIFAR-100).

The authors claim SPS+ is the first generation-based method to achieve state-of-the-art accuracy, matching or outperforming DP-SGD on CIFAR-10 and CIFAR-100 benchmarks. They also demonstrate the method's flexibility in practical applications like private federated and continual learning.

**Strengths:**

The core idea of applying data distillation techniques is a nice conceptual contribution. Their key insight is to adapt a specific family of DD—intermediate activation statistic matching (like D3S)—which is uniquely compatible with DP, as the privacy cost is incurred in a single measurement step (privatizing the statistics).

Experimental results seem strong.

**Weaknesses:**

- The presentation can be difficult to follow for a wider audience that is not already familiar with the data distillation literature. For example, I believe a more informal presentation of the pseudocode would avoid bogging down the reader with notation.
- Although it is acknowledged by the authors, the computational cost is significant and comparable to DP-SGD, which nullifies one of the advantages of DP synthetic data
- The released data statistic is very high-dimensional, as the algorithm releases both first and full second-moment (covariance) statistics. For example, SPS+ achieves SOTA classification accuracy on CIFAR10 even at $\varepsilon=1$. At this privacy level, the reported noise level $b_0 \approx 4$, while the reported dimension of the release statistic is at least $2\cdot192^2 = 73,728$, and the clipping radius is at most 3072. In other words, the magnitude of the Gaussian noise added to ensure privacy is at least $4\cdot 3072\cdot \sqrt{2*192^2} = 3,336,548$ . Since there are only 50,000 datapoints in CIFAR10, it's not clear to me if even the mean can be released with low error.

**Questions:**

- Could the authors explain how they were able to overcome the curse of dimensionality in DP?

I look forward to reading the authors' response and would be more than happy to raise my score upon clarification.

---

> ### Author Response · Authors · 2025-11-18
>
> We appreciate the reviewer’s comments and suggestions for improving the manuscript and would like to address specific concerns here.
>
> **Hard to follow pseudo-code.** We aimed to make the pseudocode very precise so that SPS could be successfully reproduced, but understandably it becomes very cumbersome to read. We will include a higher level version of the pseudocode which avoids the confusing notation and focuses more on the intuition behind the method in the rebuttal revision.
>
> **Computational cost of SPS.** As acknowledged by the reviewer, SPS currently is not significantly faster than DP-SGD. However, SPS offers other advantages. For example, SPS’ cost scales with the size of the model used for synthesis, not the model used downstream. In our case, we generated images using a smaller WRN22-8, but trained on larger models such as WRN34-10, which were unseen in generation. Future work could also look to leverage pretrained generative models in image generation to speed up convergence.
>
> **Confusion over noise scaling.** As the reviewer pointed out, as we have a much larger clipping radius, the magnitude of noise of our method is larger than that of DP-SGD, which typically uses a clipping radius of $C \approx 1- 10$. This is simply because the activation statistics we are measuring tend to have a larger magnitude than parameter gradients, which are measured in DP-SGD. We would like to point out that **the relevant factor is not the magnitude of the noise, but the signal-to-noise-ratio (SNR).** In our case, assuming dimension $d$, clipping radius $C$, noise coefficient $b_0$, and data count $N$, our measurement is sampled from a normal distribution of $\mathcal{N}(\mu, \frac{b_0 C}{N} I_d)$, where our measured statistic is $\mu$. Assuming that $|\mu| \approx C$, then the $SNR = \frac{|\mu|}{|noise|} = \frac{CN}{b_0 C \sqrt{d}} = \frac{N }{b_0 \sqrt{d}}$, **so the clipping radius does not affect the SNR, but the dimensionality does**.
>
> In our case, $ d\approx 192^2 \approx 10^5$. A WRN34-10 model has $d = 4.6 \times 10^7$ parameters, so the gradient privatized in DP-SGD is much higher dimensionality. Consequently, the **SNR of SPS can be significantly higher than direct DP-SGD.** In fact, this is one of the key advantages of SPS over DP-SGD: DP-SGD is restricted to the dimensionality of the training model, whereas SPS can tune its dimensionality by changing $D_C , D_G$. Indeed, this is why for experiments with tighter privacy budgets, we used smaller values of $D_C , D_G$ (see table 7 in appendix D).
>
> These clarifications will make their way into the main text and we will notify the reviewer once these modifications are complete.

---

### Official Review · Reviewer_ziZt · 2025-11-06

**Soundness:** 3
**Presentation:** 3
**Contribution:** 3
**Rating:** 6
**Confidence:** 3

**Summary:**

This work introduces a new algorithm for private data synthesis based on the line of dataset distillation, with a focus on image classification tasks. The key idea is to adapt dataset distillation to the privacy setting by summarizing the private dataset via intermediate activation statistics from a publicly pretrained model, privatizing these statistics, and then synthesizing images that match the privatized statistics. On CIFAR-10/100, SPS+ achieves state-of-the-art accuracy across existing DP data synthesis algorithms as well as DPSGD.

**Strengths:**

The paper is well-written. Technical writings are very clean and easy to follow.

The experiment results are very strong, SPS+ achieves accuracy competitive with, and sometimes better than, DP-SGD on CIFAR-10/100

**Weaknesses:**

The proposed SPS and SPS+ always generate a class-balanced synthetic dataset by design, without estimating or preserving the true class distribution of the private data. This is harmless for balanced benchmarks like CIFAR-10/100, but in most real-world datasets it can lead to misrepresentation of class frequencies.

The framework exposes a large number of hyperparameters, including layer selection, which has a huge search space. This will require non-trivial search and engineering effort. Furthermore, HP tuning will also incur privacy costs. The sensitivity with respect to hyperparameters needs to be discussed.

The current algorithm is restricted to image classification tasks, with all core experiments conducted on 32×32 images (CIFAR-10 and CIFAR-100). While these results are valuable and demonstrate strong performance under controlled conditions, the approach’s scalability to higher-resolution or higher-dimensional data remains unclear.

**Questions:**

See weakness.

---

> ### Author Response · Authors · 2025-11-18
>
> We appreciate the reviewer’s thoughtful comments and aim to address some of their concerns here.
>
> **Class balance assumption.** Indeed in this work we studied class-balanced datasets where the count and ratio of classes is known. This is not an uncommon assumption in DP literature [1]. We note that it is possible to modify SPS to handle class imbalance/unknown counts by additionally privately releasing class counts for each class and normalizing the means and covariances by that. Future work could study SPS’ efficacy in such settings
>
> **Hyperparameter sensitivity.** Indeed hyperparameter tuning consumes privacy budget, but in line with prior work we did not account for this cost. We will discuss this in greater detail in the rebuttal revision. As requested by the reviewer, we additionally ran experiments studying the effect of the $L_C$ (class layers) hyperparameter, and the $D_C /D_G$ hyperparameters for CIFAR-10 classification at $\epsilon = 8$ privacy budget. We summarize the results below:
>
> Table 1: SPS+ performance at CIFAR-10 classification at varying $D_C, D_G$ values. * indicates the default value used in the paper
>
> | [$D_C, D_G$]         | [256,256]  | [384,384]  | [512,512]* | [640,640]  |
> |----------------------|------------|------------|------------|------------|
> | WRN34-10             | $96.0\pm0.3$ | $96.3\pm0.2$ | $96.6\pm0.1$ | $96.4\pm0.2$ |
> | WRN 34-10 (Ensemble) | 96.7       | 96.9       | 97.2       | 97.2       |
>
> Table 2: SPS+ performance at CIFAR-10 classification at varying $L_C$ values. * indicates the default value used in the paper
> | L_C                 | [18]       | [17,18]    | [16,17,18]* | [15,16,17,18] | [14,15,16,17,18] |
> |---------------------|------------|------------|-------------|---------------|------------------|
> | WRN34-10            | $96.0\pm0.1$ | $96.3\pm0.2$ | $96.6\pm0.1$  | $96.5\pm0.2$    | $96.6\pm0.1$       |
> | WRN34-10 (Ensemble) | 96.8       | 97.1       | 97.2        | 97.2          | 97.2             |
>
>
> As we can see, performance varies very little with respect to these hyperparameters. Due to the fixed privacy budget, the noise added to each parameter scales inversely with the parameter count: for larger $L_C$ or $D_C/D_G$ values, the noise added is larger, compensating for receiving more statistics about the dataset. Consequently, SPS’ performance is generally stable with respect to these hyperparameters.
>
> **Small datasets.** We focused on lower resolution data such as CIFAR-10, and CIFAR-100. This was mainly due to the constraint of requiring a larger public dataset for pretraining that is disjoint from the private dataset. [1,3] are able to leverage proprietary datasets such as JFT-300M/JFT-4B for their pretraining datasets, which we do not have access to. Nonetheless, in our work we ran the CAMLEYON-17 experiment at 64x64. Furthermore, **we ran additional experiments on Tiny-ImageNet (64x64, 100k samples, 200 classes) at $\epsilon=8$ with SPS+**. For the pretraining dataset we used a 64x64 variant of ImageNet-1K, with overlapping classes with Tiny-ImageNet removed, leading to a 818-class pretraining dataset. At $\epsilon = 8, \delta = 4e-6$, we achieve $49.5\%$ accuracy with an ensemble of WRN28-10 models, using a WRN22-8 model for distillation.
>
> **Other contributions.** We would also like to point out other contributions of this work which were not addressed by the reviewer, namely SPS’s application to federated learning and continual learning. SPS adapts without modification to both settings, whereas DP-SGD, for example, fails in continual learning due to the catastrophic forgetting phenomenon.
>
> These new results will make their way into the main text and we will notify the reviewer once these modifications are complete.
>
> [1] Differentially Private Image Classification from Features
>
> [2] CoinPress: Practical Private Mean and Covariance Estimation
>
> [3] Unlocking High-Accuracy Differentially Private Image Classification through Scale
>
> [4] Differentially Private Synthetic Data via Foundation Model APIs 1: Images

---

### Author Response · Authors · 2025-12-03
**Rebuttal Revision**

Here we document specific changes to the manuscript made during the revision to address reviewer's concerns:

1. **Additional Hyperparameter Sensitivity Experiments** in appendix B.5. to address Reviewer ziZt's concern about the additional hyperparameters exposed by our method. We see that our method is robust to hyperparameter configurations

2. **Experiments of SPS+ synthesizing datasets larger than the source dataset** in section 5.4 table 3 and appendix table 14 and 15 as requested by Reviewer UvM8. More performance can be attained by generating even larger datasets.

3. **Additional Experiments at larger scale datasets** in appendix G.1, as requested by Reviewer ziZt which show that SPS+ can be used effectively on Tiny-ImageNet using a public dataset of ImageNet-1K with overlapping classes removed.

4. **Additional Comparison of SPS+ in terms of visual fidelity** in section 5.4 figure 3,5 and appendix table 13 as requested by Reviewer UvM8.

5. **More detailed description of Federated learning setting** in section 5.5 and discussing possible future work to overcome non-IID settings in Appendix B.6, as requested by Reviewer UvM8.

6. **More detail how SPS can tune its dimensionality to increase SNR** in section 3.2.2 and lines 239-242, to clarify points raised by Reviewer fbnc

We hope that these change address the reviewer's concerns accordingly.

---

### Meta-Review · Area_Chair_ez1U · 2025-12-18

**Summary:**

The paper presents a new DP data distillation method based on synthesizing data that match global and class-specific first and second order statistics in random projections of the feature space of a pre-trained model, and then fine-tuning using these data.

The original reviews are mixed on the borderline. The reviewers note the following weaknesses:
1. The method is restricted to balanced classification tasks.
2. Many hyperparameters, unclear how they are handled.
3. Restricted to image classification with demonstrations only with small images.
4. Lack of accessible presentation of the algorithm
5. High computational cost.
6. Unclear how method handles the high noise required for low $\varepsilon$.
7. Lack of comparison with other data synthesis methods.

I would add the following weaknesses:
8. Lack of evaluation of DP-SGD with parameter-efficient fine-tuning (PEFT; e.g. LoRA), which has been shown to be highly effective e.g. for CIFAR-100.
9. Limited set of evaluation datasets, limited to ones where last layer fine-tuning works well (in comparison to full fine-tuning), which does not provide information on how the method would perform outside this range.

To clarify the last two points: the method is clearly targeted to a setting where classes are (approximately) linearly separable in the feature space of the pre-trained model. In this setting, fine-tuning the last layer is often effective and computationally extremely cheap, because the outputs from the backbone can be cached.

What I am missing from the paper is an honest evaluation of the compute-accuracy trade-off with two obvious baselines: computationally cheap last-layer fine-tuning and more accurate PEFT. In case of CIFAR-100, PEFT can give 10%-point boost to classification accuracy, so quoting the non-PEFT De et al. accuracy as SotA without context is highly misleading. (See https://arxiv.org/abs/2302.01190 for a comparison although with different pre-training.)

**Reviewer Concerns:**

The authors provide partial responses to weaknesses #1-#3 in their response. For #4 they promise to add new pseudocode, but it is not obvious where this is in the revised manuscript. The authors provide clarifications regarding #5 and #6 and highlight additional results to address #7.

Overall, the authors seem to have addressed all the original reviewer comments except for #4 where I was unable to find the new pseudocode.

The new weaknesses #8 and #9 are obviously not addressed, and given the paper's positioning itself as a DP-SGD replacement, I believe these would be important to address.

**Reviewer Scores:**

I find it hard to judge how the reviewers would have reacted, but given the limited evidence that the paper provides for the claims of being better DP-SGD as discussed above, I feel that it cannot be accepted to ICLR.

Finally, I really appreciate the work and believe that a data distillation approach can have unique advantages compared to DP-SGD for example in continual and federated learning as also noted by the authors, but I feel that the way the results are presented as a generic replacement to DP-SGD could be misleading and requires further supporting evidence, or alternatively a re-framing of the paper more towards data distillation.

---

### Decision · Program_Chairs · 2026-01-26

Reject